# BOOTSTRAPPING LANGUAGE AND NUMERICAL FEEDBACK FOR REINFORCEMENT LEARNING IN LLMS

## ABSTRACT

Reinforcement learning for large language models (LLMs) often relies on scalar rewards, a practice that discards valuable textual rationale buried in the rollouts and hampers training efficiency. Naive attempts to incorporate language feedback are often counterproductive, risking either memorization from leaked solutions or policy collapse from irrelevant context. To address this, we propose **Language-And-Numerical Policy Optimization (LANPO)**, a framework that cleanly separates the roles of feedback: language guides exploration, while numerical rewards drive optimization. LANPO builds a dynamic experience pool from past trials and introduces two principles to ensure feedback is effective: *Reward-Agnostic Reflection* for safe intra-sample self-correction and *Relevant Abstraction* to distill generalizable lessons from inter-sample experiences. Across mathematical reasoning benchmarks, LANPO enables 7B and 14B models to significantly outperform strong baselines trained with GRPO in test accuracy. Our work provides a robust method for integrating historical experiences into the LLM RL loop, creating more effective and data-efficient learning agents.

## 1 INTRODUCTION

Reinforcement learning (RL) has become a central ingredient for improving the reasoning abilities of large language models (LLMs) OpenAI (2024); Guo et al. (2025). In the prevalent pipeline, a model's complex reasoning is assessed by a programmatic verifier or an LLM judge, which compresses its evaluation into a single scalar reward. Policy optimization algorithms like PPO or its variants then update the model's parameters to maximize this scalar signal (Schulman et al., 2017; Shao et al., 2024a). While effective, this scalarization of feedback discards the rich, explanatory rationale hidden in the model's textual responses. Consequently, exploration proceeds largely de novo for each prompt; the model cannot explicitly reason about why a previous attempt failed and must generate new rollouts without reusing these lesson-like experiences. This leads to repetitive, low-diversity exploration where failure patterns persist, causing state-of-the-art reasoning models to require thousands of RL steps to train He et al. (2025).

Unlike conventional RL agents, LLMs possess the unique ability to process and generate nuanced language feedback (Brown et al., 2020). This opens the door to learning from past trials by retrieving relevant knowledge or reasoning templates within the context window (Lewis et al., 2021; Yang et al., 2024b). However, naively integrating language feedback into the RL training loop introduces a fundamental paradox. On one hand, providing feedback from trials on the *same* problem (intra-sample feedback) risks **information leakage**; the model may learn to simply copy the correct answer, inflating training performance while undermining generalization. On the other hand, using feedback from *different* problems (inter-sample feedback) often leads to **behavior collapse**, where the model ignores the provided context as it is often too specific or irrelevant, finding it easier to generate a solution from scratch. This dilemma has left language feedback as an underutilized resource in mainstream LLM training.

To resolve this tension, we propose **Language-And-Numerical Policy Optimization (LANPO)**, a training paradigm that synergistically *bootstraps language and numerical feedback* to enhance learning efficiency. As depicted in Figure 1 (right), LANPO unifies these two signals: language feedback is used to guide and enrich exploration via context updates, while numerical rewards are retained to drive robust policy optimization through parameter updates. At its core, LANPO introduces an *experience pool* that accumulates and distills past trials into concise, reusable natural-language

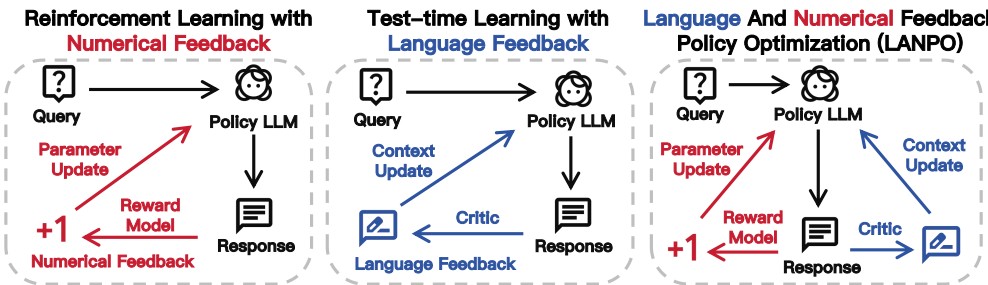

Figure 1: Comparison between three learning paradigms for LLMs. **Left:** RL with numerical feedback adopts scalar rewards as the primary source of guidance for learning, where the actor cannot explicitly learn from past experiences. **Middle:** Test-time learning with language feedback features the LLM's ability to learn and adapt within its context window without parameter updates. **Right:** Our proposed language and numerical policy optimization is an RL algorithm that unifies the two by extracting meaningful language feedback from the previously discarded rollouts.

summaries. To prevent the pitfalls of naive integration, we introduce two key mechanisms: (1) **Reward-Agnostic Reflection** for intra-sample feedback, where the model critiques and refines its own past attempts without access to the ground truth, thereby preventing leakage. (2) **Relevant Abstraction** for inter-sample feedback, which filters for semantically similar problems and summarizes their solutions into high-level principles, ensuring the guidance is both useful and generalizable, thus avoiding behavior collapse.

To summarize, our contributions are threefold: (1) **Identification and Mitigation of Core Failure Modes:** We identify and analyze two critical failure modes—information leakage and behavior collapse—that impede the effective integration of language feedback within Reinforcement Learning frameworks for LLMs. To address these challenges, we introduce two novel techniques, Reward-Agnostic Reflection and Relevant Abstraction, which are designed to safely and effectively extract valuable information from training rollouts. (2) **A Robust Implementation Framework:** We present LANPO, a practical framework that operationalizes our proposed techniques. LANPO consists of three core components: an experience pool, a multi-role LLM actor, and a mixture-of-modes training schedule. Together, these components enhance the robustness and versatility of the hybrid language-numerical learning paradigm. (3) **Empirical Validation of Effectiveness:** We conduct an extensive empirical evaluation on challenging mathematical reasoning benchmarks. Our results demonstrate that LANPO consistently outperforms the strong GRPO baseline in sample efficiency. Notably, LANPO achieves an absolute performance improvement of up to 9.27% on the AIME25 test set after the same number of training steps.

## 2 RELATED WORK

Our work, LANPO, builds upon and intersects with three primary areas of research: Reinforcement Learning (RL) for LLMs, the use of language feedback for model improvement, and memory-augmented agent architectures.

**RL with Numerical Feedback.** The practice of optimizing LLMs with scalar rewards has become a cornerstone of developing advanced models (Ziegler et al., 2019), leading to powerful instruction-following agents (Ouyang et al., 2022; Bai et al., 2022) and specialized problem-solvers (OpenAI, 2024; Guo et al., 2025). The underlying algorithms have also evolved from Proximal Policy Optimization (PPO) (Schulman et al., 2017) to more recent methods like Direct Preference Optimization (DPO) (Rafailov et al., 2023) and GRPO (Shao et al., 2024a). Our work is orthogonal to the choice of the specific optimization algorithm. LANPO operates a level above, introducing a language-feedback layer that structures the context provided to the policy. This layer is designed to improve the quality and efficiency of exploration before the numerical reward is used for the policy update, making it a complementary component to any of these RL frameworks.

**Language Feedback at Test Time.** Using language to refine model outputs is a well-explored area. This includes inference-time correction (Wang et al., 2024a; Kumar et al., 2024), generating self-critiques (Madaan et al., 2023; Yuan et al., 2024; Ankner et al., 2024), and maintaining reflections

across episodes (Shinn et al., 2023; Yuan & Xie, 2025). Other works use feedback as in-context examples to guide generation (Chen et al., 2024; Baronio et al., 2025; Chen et al., 2025; Li et al., 2025). LANPO's contribution lies in how it systematically integrates language feedback into the RL training loop to overcome specific failure modes. Our **reward-agnostic reflection** for intra-sample feedback differs from prior self-correction methods by being fully integrated into a single-turn RL process without access to gold labels, thus preventing information leakage. For inter-sample feedback, our **relevant abstraction** mechanism—which filters and summarizes trajectories into transferable principles—directly counteracts the behavior collapse that can occur when naively reusing raw solutions as context.

**Memory-Augmented Language Agents.** The concept of an external memory to store and reuse past experiences is central to many advanced agents. These memories have been used to build skill libraries (Wang et al., 2023; Yang et al., 2025b), correct errors post-deployment (Madaan et al., 2022), and serve as an episodic "Case Bank" at test time (Wang et al., 2025). While LANPO's experience pool serves a similar function, it is uniquely designed for the RL training loop. Rather than just storing raw trajectories for retrieval, LANPO actively processes on-policy rollouts into abstracted summaries of "principles and pitfalls." This distilled knowledge becomes a direct input for shaping exploration in subsequent RL episodes, creating a tight, synergistic loop between experience, exploration, and optimization that is absent in architectures where memory is primarily a test-time or inference-time resource.

## 3  CHALLENGES IN INTRODUCING LANGUAGE FEEDBACK

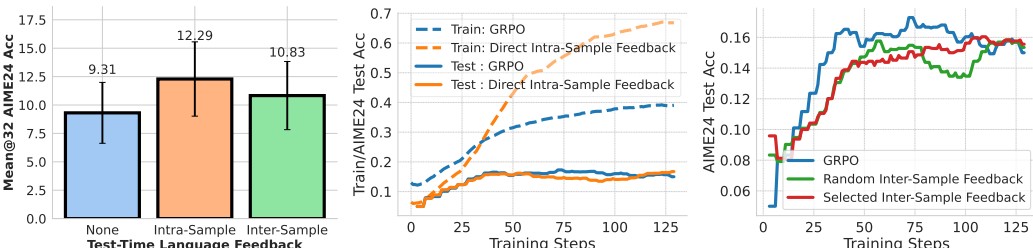

(a) Test-time language feedback.  (b) RL with intra-sample feedback.  (c) RL with inter-sample feedback.

Figure 2: **Challenges in introducing language feedback to RL training.** (a) At test time, both intra-sample feedback (self-correction) and inter-sample feedback (in-context examples) yield clear accuracy gains without sophisticated design. (b) However, intra-sample feedback in training suffers from **information leakage**: when the actor can access the ground-truth answer to the exact problem, training accuracy rises sharply but fails to translate into test-time improvement. (c) Inter-sample feedback in training, where correct rollouts are reused across problems, fails to surpass GRPO and often induces **behavior collapse**, in contrast to the clear benefits seen in inference time (9.31 → 10.83 from (a).)

Large language models trained with RL typically generate thousands of rollouts per iteration, which are then discarded after reward estimation. Yet, these rollouts contain rich intermediate reasoning steps and successful solution trajectories that could, in principle, serve as *language feedback* to guide exploration more effectively. If feedback that improves accuracy at test time could be incorporated during training, it might accelerate policy search and unlock progress on harder tasks.

We therefore begin by revisiting the effectiveness of language feedback in inference. Using Qwen2.5-7B-Instruct (Yang et al., 2024a) on the AIME24 benchmark, we find that both *intra-sample feedback* (self-correction on wrong attempts (Madaan et al., 2023; Wang et al., 2024b)) and *inter-sample feedback* (in-context examples retrieved from MATH500 (Brown et al., 2020)) each boost mean@32 accuracy compared to the baseline (Figure 2a).

This confirms that language feedback is indeed useful at test time, motivating us to explore incorporating these language feedback into RL training. However, our preliminary experiments reveal two critical obstacles when naively applying language feedback in training:

**Intra-sample feedback risks information leakage.** We provided the correct answer to the same training problem during rollouts, akin to rejection fine-tuning (RFT). As shown in Figure 2b, training accuracy (mean@8) spikes quickly, but test accuracy shows no improvement over GRPO. The model

learns to exploit the leaked labels rather than genuinely improve its reasoning ability. Moreover, at inference time, there is no oracle to indicate which solution to correct, making this strategy infeasible without a label-free design.

**Inter-sample feedback suffers from behavior collapse.** We attempted to reuse correct trajectories discovered during training as in-context demonstrations for other problems, either by random sampling or by selecting similar problems. Both strategies, shown in Figure 2c, fail to outperform GRPO. Closer inspection reveals that the model often ignores the provided examples and directly outputs answers, a phenomenon we refer as *behavior collapse*. This starkly contrasts with the effectiveness of ICL at inference, underscoring a disconnect between test-time and training-time dynamics. We present further discussion into behavior collapse in Appendix A.1.

In summary, while language feedback has clear potential to accelerate exploration in RL, naive integration during training introduces pitfalls: intra-sample feedback leaks labels and leads to overfitting, while inter-sample feedback collapses into ineffective behavior. These challenges motivate the need for principled strategies to design language feedback mechanisms that can genuinely improve RL training—a direction we pursue in the next section.

# 4 LANPO: LANGUAGE-AND-NUMERICAL POLICY OPTIMIZATION

Our preliminary study (Section 3) exposed a paradox: while language feedback improves test-time accuracy, naïve attempts to integrate it into RL training suffers from *information leakage* or *behavior collapse*. This motivates the design of **Language-And-Numerical Policy Optimization (LANPO)**, which introduces mechanisms that allow language and numerical feedback to *bootstrap one another*. Language feedback accelerates RL exploration by reusing knowledge from past trajectories reflection, while numerical rewards identify and reinforce the valuable ones, yielding a stronger policy that in turn generates better feedback. Through this mutual reinforcement, LANPO transforms signals that previously conflicted into complementary drivers of efficient and robust policy learning.

## 4.1 METHODOLOGY

Our preliminary study revealed two major obstacles to using language feedback in RL: *information leakage* in intra-sample feedback and *behavior collapse* in inter-sample feedback. LANPO addresses these pitfalls with two key mechanisms.

**Reward-agnostic reflection for Intra-sample Feedback.** Naïve intra-sample feedback, where the gold solution is revealed, inflates training accuracy but undermines generalization by encouraging memorization of leaked labels. LANPO replaces this with a *reward-agnostic reflection* mechanism. Instead of accessing the true label, the model revisits its own earlier attempts, critiques them step by step, and then produces a refined solution. This encourages reflective exploration without exposing correctness signals. Unlike prior multi-stage self-correction methods (Kumar et al., 2024), our approach integrates seamlessly into single-turn RL training, treating past attempts as structured context rather than hidden supervision.

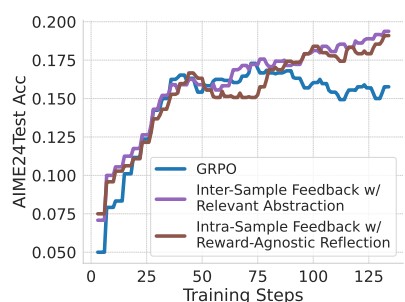

Figure 3: Effectiveness of our algorithm design: For inter-sample feedback, we conduct similarity based selection and high-level summarization. Meanwhile, we adopt self-reflection to review intra-sample feedback and explore based on past attempts.

**Relevant Abstraction for Inter-sample Feedback.** For inter-sample feedback, naïvely reusing raw solutions often triggers behavior collapse: the model learns to ignore the provided context and instead answers directly, since this path is equally rewarded and usually simpler. LANPO overcomes this by introducing *relevant abstraction*, which ensures that reused experiences are both semantically aligned with the current problem and distilled into transferable knowledge. The process has three steps. First, *similarity-based filtering* restricts retrieval to trajectories drawn from sufficiently related problems, guaranteeing that the added context is more useful than starting from scratch. Second, *summarization and abstraction* condense raw solutions into high-level principles

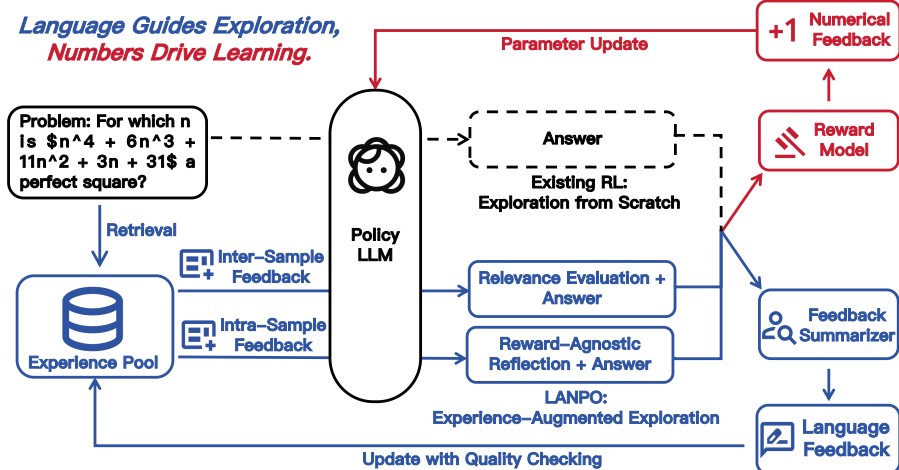

Figure 4: **LANPO training pipeline.** The Language-Guided Exploration Loop shapes the policy's exploration. An experience pool stores abstracted principles from past trajectories. The actor uses this pool to perform inter-sample exploration (using guidance from related problems) and intra-sample reflection (critiquing its own past attempts). Successful solutions are summarized and added back to the pool. The Numerical-Driven Learning Loop optimizes the policy. All attempts, whether guided by language or not, are evaluated by a Reward Model. The numerical reward is then used to update the policy via reinforcement learning. This architecture cleanly separates the roles of feedback: language guides where to explore, while numerical rewards determine what to learn.

and common pitfalls that can generalize across problems, rather than problem-specific steps. Third, the actor is explicitly instructed to analyze the retrieved feedback before producing its own plan and solution, reinforcing active engagement with the context. As shown in Figure 3, relevant abstraction makes inter-sample feedback consistently beneficial, avoiding collapse and yielding substantial gains over naïve reuse of rollouts.

Together, reward-agnostic reflection and relevant abstraction transform the pitfalls of language feedback into guiding principles for exploration. Intra-sample feedback fosters self-reflection without leakage, while inter-sample feedback provides reliable, transferable hints without collapse.

## 4.2 LANPO MODULES AND TRAINING OBJECTIVE

Building on the principles of *Reward-Agnostic Reflection* and *Relevant Abstraction*, LANPO organizes training around a modular pipeline. The design brings these ideas to life through a shared experience pool, two specialized responders, and a stable on-policy objective.

**Experience pool.** At the center of LANPO is a capped-size experience pool $\mathcal{E}$ that accumulates distilled experiences from past rollouts. Each entry contains a structured summary with a *flow of thought*, transferable *principles and pitfalls*, and metadata such as reward, source, and timestamp. When solving a new problem $x$, the policy retrieves context $c \sim p_c(\cdot \mid x, \mathcal{E})$: recent attempts on the same $x$ provide material for reflection, while filtered entries from semantically related problems supply abstracted guidance. In this way, the pool serves as a dynamic memory that fuels both reflection and abstraction.

**Inter-sample exploration.** To make retrieved context genuinely useful rather than ignored, LANPO equips the policy with an inter-sample exploration module. This component consumes summarized successful solutions from related problems and evaluates its relevance with the problem at hand. As discussed earlier, only experiences with relevance higher than $\beta$ will be incorporated into the policy's context. In doing so, it enforces the practice of drawing on transferable principles instead of copying raw solutions, directly realizing the idea of Relevant Abstraction. Summarization and filtering ensure that the retrieved feedback is both relevant and generalizable, while the exploration process encourages active engagement with the context during reasoning.

**Intra-sample exploration.** In parallel, LANPO introduces an intra-sample exploration module, which revisits the model's own earlier attempts, critiques them step by step, and refines the reasoning into a revised solution. This operationalizes Reward-Agnostic Reflection: the model learns to improve on its own outputs without ever relying on gold labels, fostering a self-corrective habit that strengthens exploration while avoiding leakage. Because this process is integrated into single-turn RL training, reflection arises naturally within the rollout itself rather than requiring a separate stage.

**Seeding atomic capabilities via SFT.** Since neither summarization nor feedback-driven reasoning is innate to a base LLM, LANPO begins with a lightweight supervised fine-tuning (SFT) stage to provide the policy with three atomic skills: (i) a *Summarizer* that converts raw trajectories into concise entries for the pool, (ii) an *inter-sample exploration* capability that learns to evaluate and apply retrieved feedback, and (iii) an *intra-sample exploration* capability that learns to audit and refine past attempts. The purpose of this stage is not to maximize accuracy, but to instill literacy—the ability to write, read, and act on structured feedback—so that RL training can fully exploit the principles of reflection and abstraction.

**RL Training loop and objective.** During RL, the model alternates between feedback-aware roll-outs and from-scratch rollouts. With probability $p_t$, a context $c$ is drawn from $\mathcal{E}$, activating either reflection or abstraction; with probability $1-p_t$, $c = \emptyset$, preserving the ability to reason without additional context. New trajectories are summarized and added back into the pool, gradually improving its quality. The policy is updated using a GRPO-style surrogate objective. Our key modification is to introduce an expectation over the context $c$, which is sampled from the language experience pool $\mathcal{E}$. This makes the standard PPO objective feedback-aware. The full objective is:

$$\mathcal{L}_{\text{LANPO}}(\theta) = \hat{\mathbb{E}}_t \left[ \mathbb{E}_{c \sim p_c(\cdot|x_t, \mathcal{E})} \left[ \mathcal{L}^{\text{CLIP}}(\theta, c) \right] \right] \tag{1}$$

where $\mathcal{L}^{\text{CLIP}}(\theta, c)$ is the context-dependent clipped surrogate objective from PPO:

$$\mathcal{L}^{\text{CLIP}}(\theta, c) = \min\left( r_t(\theta, c)\hat{A}_t, \ \text{clip}(r_t(\theta, c), 1-\epsilon, 1+\epsilon)\hat{A}_t \right). \tag{2}$$

The probability ratio $r_t(\theta, c) = \frac{\pi_\theta(y_t|x_t, c)}{\pi_{\theta_{\text{old}}}(y_t|x_t, c)}$ compares the new policy to the old one used for data collection, conditioned on the sampled context $c$.

Here, $\hat{A}_t = (r - \hat{b}(x_t))/\hat{\sigma}(x_t)$ is the estimated advantage at timestep $t$, and $\epsilon$ is the clipping hyper-parameter. This formulation clearly shows that LANPO is an orthogonal enhancement to PPO-style algorithms: we do not alter the core clipping mechanism but rather guide the policy optimization by averaging the objective over a distribution of relevant language feedback. The retrieval distribution $p_c$ encodes how exploration is shaped by this feedback.

## 4.3 EXPERIENCE-DRIVEN INFERENCE AT TEST TIME

Finally, the same mechanisms that enabled stable training also make LANPO models natively *experience-driven* at inference. At test time, the policy can:

- **Solve from scratch** with no external context, preserving robustness and efficiency.
- **Retrieve experience** by (i) retrieving relevant entries from the final experience pool $\mathcal{E}$ for inter-sample guidance, or (ii) applying the intra-sample reflection loop on its own first attempt.

Because the experience pool contains distilled, transferable lessons rather than raw solutions, and because the self-reflection mechanism we designed is label-free, these inference modes has the potential to improve performance without external input.

## 5 EXPERIMENTS

We now turn to a detailed empirical study of LANPO. Our experiments examine its overall effectiveness, the roles of different design components, and the training dynamics that shed light on why it works. Each analysis connects back to the challenges identified in Section 3. We briefly summarize our setup below and defer full details, including hyperparameters, to Appendix B.2.

Table 1: Performance comparison of Qwen models with different RL strategies and inference modes. The highest score for each metric within each model group is highlighted in **bold**.

| Model | Training Method | Inference Mode | AIME 25 | AIME 24 | AMC | MATH | Avg |
|---|---|---|---|---|---|---|---|
| Qwen2.5-7B-Base-SFT | No RL | Zero-shot | 7.60 | 10.83 | 39.57 | 70.60 | 32.15 |
| | GRPO | Zero-shot | 13.02 | 17.29 | 52.11 | 79.80 | 40.56 |
| | LANPO w/ Intra | Zero-shot | 13.96 | 20.42 | 60.50 | 82.60 | 44.37 |
| | | w/ Self-correction | 15.73 | **22.19** | 62.54 | 82.60 | 45.77 |
| | LANPO w/ Inter | Zero-shot | 16.04 | 19.48 | 59.83 | 81.20 | 44.14 |
| | | w/ Retrieval | 16.98 | 19.48 | 63.06 | 79.80 | 44.83 |
| | LANPO w/ Both | Zero-shot | 16.77 | 20.52 | 59.71 | 82.60 | 44.90 |
| | | w/ Retrieval | 16.04 | 21.35 | **63.55** | 82.00 | 45.74 |
| | | w/ Self-correction | **17.71** | **22.19** | **63.55** | **83.20** | **46.66** |
| Qwen3-14B-Base-SFT | No RL | Zero-shot | 18.12 | 20.73 | 58.43 | 85.20 | 45.62 |
| | GRPO | Zero-shot | 33.02 | 47.40 | 78.20 | 92.00 | 62.66 |
| | LANPO w/ Intra | Zero-shot | 38.23 | 46.15 | 79.89 | 91.80 | 64.02 |
| | | w/ Self-correction | **42.29** | **53.75** | 82.49 | 92.80 | **67.83** |
| | LANPO w/ Inter | Zero-shot | 36.88 | 48.65 | 81.70 | 92.40 | 64.91 |
| | | w/ Retrieval | 35.62 | 46.77 | 80.80 | 91.40 | 63.65 |
| | LANPO w/ Both | Zero-shot | 34.17 | 43.23 | 81.70 | 92.40 | 62.88 |
| | | w/ Retrieval | 34.06 | 43.33 | 78.84 | 90.60 | 61.71 |
| | | w/ Self-correction | 37.50 | 48.65 | **82.53** | **93.60** | 65.54 |

**Models and datasets.** We evaluate two base models: Qwen2.5-7B (Yang et al., 2024a) and Qwen3-14B (Yang et al., 2025a), neither of which are instruction-tuned. For RL training, we use the DAPO dataset (Yu et al., 2025), which contains ∼17K competition-level math problems. Performance is measured on AIME25, AIME24, AMC23, and MATH-500 (Hendrycks et al., 2021) with mean@32 accuracy following common evaluation configurations.

**Training protocols.** RL training is performed with VeRL framework (Sheng et al., 2024), using GRPO (Shao et al., 2024b) with group size 16 as the policy loss. We adopt the clip-higher trick (Yu et al., 2025) with $\epsilon_{low} = 0.2$, $\epsilon_{high} = 0.28$. Prompts are truncated at 3,072 tokens and generations at 8,192 tokens. Lastly, the RL training steps is set to 330 (10 epochs) by default.

**LANPO configuration.** LANPO pre-computes relatedness scores using Qwen2.5-7B-Math and retains only experiences with similarity $\geq 0.9$, details for which is presented in Appendix C.1. Unless otherwise stated, the feedback ratio $p_t$ is 0.5. An initial SFT stage with 3K QA pairs from DeepSeek-V3 DeepSeek-AI et al. (2025) equips models with summarization, inspection, and response skills before RL training. We enable inter-sample feedback at test time by retrieving from the experience pool accumulated by RL training. Meanwhile, benefiting from our reward-agnostic design of intra-sample feedback, the actor after training is able to self-correct with a two-turn conversation without external hints Kumar et al. (2024).

## 5.1 BENCHMARK PERFORMANCE

**Zero-Shot Inference Performance.** Table 1 presents the main comparison between GRPO and LANPO. On Qwen2.5-7B, GRPO achieves an average accuracy of 40.56, whereas LANPO consistently pushes this higher: intra-sample feedback reaches 44.37, inter-sample feedback 44.14, and

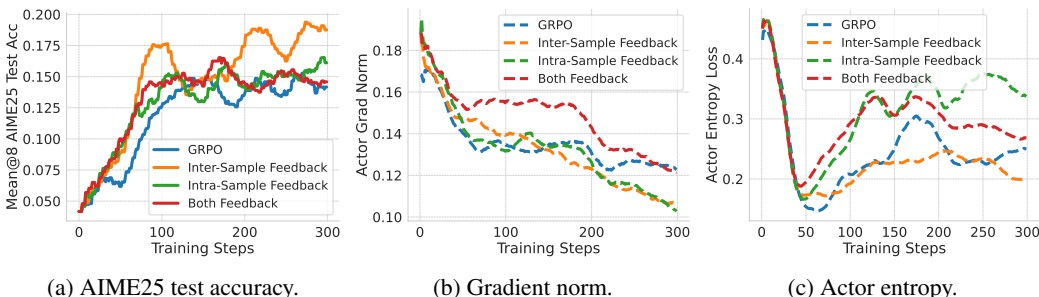

(a) AIME25 test accuracy.    (b) Gradient norm.    (c) Actor entropy.

Figure 5: **Training dynamics.** We plot key metrics for LANPO variants and the GRPO baseline on Qwen2.5-7B. (a) Mean@8 test accuracy on the AIME25 benchmark, where both feedback mechanisms improve performance throughout training process. (b) Actor gradient norm, showing stable optimization of LANPO. (c) Policy Entropy: The feedback mechanisms have distinct effects on exploration. Intra-sample feedback maintains high entropy, while inter-sample feedback reduces entropy.

Table 2: **Necessity of relevance filtering for inter-sample feedback.** The results demonstrate that without a proper filtering mechanism, providing inter-sample feedback can be detrimental to performance, as seen in the performance drops for GRPO and naive retrieval. By applying a relevance filter ($\gamma = 0.9$), we reverse this trend and achieve the best overall results.

| Qwen2.5-7B-Base | Inference Mode | AIME 25 | AIME 24 | AMC | MATH | Avg |
|---|---|---|---|---|---|---|
| GRPO | Zero-shot | 13.02 | 17.29 | 52.11 | 79.80 | 40.56 |
| | w/ Retrieval | 12.50 | 16.25 | 51.17 | 76.60 | 39.13 |
| w/o filtering ($\gamma = 0.0$) | Zero-shot | 15.21 | **19.69** | 60.66 | **83.00** | 44.64 |
| | w/ Retrieval | 13.65 | 18.65 | 62.73 | 81.40 | 44.11 |
| w/ filtering ($\gamma = 0.9$ ) | Zero-shot | 16.04 | 19.48 | 59.83 | 81.20 | 44.14 |
| | w/ Retrieval | **16.98** | 19.48 | **63.06** | 79.80 | **44.83** |

combining both with self-correction yields 46.66. The gains extend to Qwen3-14B as well, where LANPO with inter-sample feedback improves the average accuracy to 64.91 compared to 62.66 for GRPO.

These improvements demonstrate that LANPO systematically enhances the policy's ability to solve problems without additional hints at test time. By making feedback leakage-free and collapse-resistant during training, the resulting policy internalizes more reusable reasoning strategies, which translates into stronger zero-shot performance.

**Experience Augmented Inference.** A second question is whether LANPO-trained models can also benefit from explicit language feedback at inference. The answer we observe is yes. As Table 1 shows, adding a self-correction step further boosts Qwen2.5-7B from 44.90 (zero-shot with both feedback pathways) to 46.66. Similarly, Qwen3-14B with intra-sample feedback improves from 64.02 to 67.83 when allowed to self-correct. Retrieval also helps in selective cases, particularly when similarity filtering ensures relevance.

This confirms that LANPO not only yields stronger stand-alone solvers but also equips them with the ability to reuse experiences dynamically at test time. In other words, the experience-driven inference behavior that motivated our method emerges naturally as a byproduct of training.

## 5.2 EMPIRICAL UNDERSTANDINGS

**Training Dynamics.** To understand how LANPO alters the learning process, we inspect the training dynamics on the AIME25 benchmark, as illustrated in figure 5. Both intra- and inter-sample feedback mechanisms clearly outperform the GRPO baseline, achieving faster convergence and a higher final test accuracy (figure 5a). These performance gains are realized without compromising stability, as evidenced by the smooth decline in the actor gradient norm for all methods (figure 5b).

Table 3: **Influence of the feedback ratio $(p_t)$ on model performance.** A moderate ratio of $p_t = 0.50$ achieves the optimal balance between leveraging past experiences and preserving generalization, resulting in the highest average accuracy.

| Feedback Ratio $(p_t)$ | Inference Mode | AIME 25 | AIME 24 | AMC | MATH | Avg |
|---|---|---|---|---|---|---|
| GRPO $(p_t = 0.0)$ | zero-shot | 13.02 | 17.29 | 52.11 | 79.80 | 40.56 |
| | w/ self-correction | 13.85 | 18.02 | 53.24 | 80.40 | 41.38 |
| $p_t = 0.25$ | zero-shot | 15.10 | 20.31 | 57.68 | 78.80 | 42.97 |
| | w/ self-correction | 16.67 | 20.21 | 58.36 | 80.60 | 43.96 |
| $p_t = 0.50$ | zero-shot | 13.96 | 20.42 | 60.50 | **82.60** | 44.37 |
| | w/ self-correction | 15.73 | **22.19** | **62.54** | **82.60** | **45.77** |
| $p_t = 0.75$ | zero-shot | 17.71 | 17.29 | 58.02 | 78.40 | 42.86 |
| | w/ self-correction | **19.06** | 18.85 | 58.62 | 80.20 | 44.18 |

The actor policy entropy in figure 5c reveals how our feedback mechanisms distinctly shape exploration. Intra-sample feedback, driven by *Reward-Agnostic Reflection*, sustains the highest entropy. This suggests the policy is encouraged to critique its own attempts and deviate from familiar patterns, thereby broadening its exploration. Conversely, inter-sample feedback, guided by *Relevant Abstraction*, produces the lowest entropy. This indicates that providing targeted principles effectively prunes the search space, focusing the policy's exploration. In summary, these dynamics confirm that LANPO effectively uses language to guide the learning process: reflection prevents the policy from repeating familiar strategies, while abstraction provides strong heuristics to accelerate progress. We provide more visualization of training curves in Appendix A.3.

**Ablation: Necessity of Filtering.** We next ablate the inter-sample feedback filtering mechanism. Table 2 shows that both GRPO and naïve retrieval without filtering can even suffer performance decrease when provided with experience, with GRPO decreasing for 1.43 and naive retrieval dropping for 0.53. In contrast, filtering at $\gamma = 0.9$ reverses this effect, raising performance to 44.83. This demonstrates that collapse-resistant design is not optional: without proper filtering, the model cannot learn to effectively use inter-sample feedback at test time. We note that the key take-aways here should not be the specific magnitude of this gain, but the fact that our method successfully unlocks this mode of reasoning, which is difficult with naive training

**Ablation: Effect of Feedback Ratio.** Table 3 proceeds by studying the probability $p_t$ of including feedback during LANPO training. We adopt intra-sample feedback for this ablation. The results reveal that a low value (0.25) underutilize past experiences, while very high value (0.75) results in the lowest zero-shot performance on average (42.86). The best trade-off arises at moderate ratios: $p_t = 0.50$ achieves the highest average accuracy (45.77 with self-correction), outperforming both lower and higher ratios. This confirms that balancing feedback-aware and from-scratch rollouts is key to preserving generalization.

**Exploration with Language Feedback.** Finally, we analyze how language feedback reshapes exploration. Models trained with LANPO is able to produce rich reasoning chains that reference retrieved principles or critically examine their own prior attempts, which is supported by the examples provided in Appendix D.

## 6 DISCUSSION AND CONCLUSION

**Remark on training overhead:** LANPO introduces computational overhead from three primary sources: (1) managing the experience pool, (2) processing longer sequences for feedback-guided rollouts, and (3) an auxiliary summarization stage. The cost of experience pool management (storage, filtering, and retrieval) is minimal, as these operations are lightweight and can be executed on CPUs. The primary overhead arises from increased sequence lengths. Both inter/intra-sample expand the model's input prompt and generated output. Since transformer computation scales superlinearly with sequence length, this directly increases the processing time per step. Additionally, the summarization step requires a separate generation process. In all, LANPO's primary costs stem from processing additional tokens during rollouts and summarization. These can be substantially mitigated with standard acceleration techniques, such as asynchronous RL frameworks (Wu et al., 2025) to hide latency or model quantization to speed up forward passes (Krishnan et al., 2022).

To summarize, this work demonstrates that the rollouts generated during LLM RL training can be harnessed as language feedback to reliably improve sample efficiency when paired with safeguards that prevent leakage and collapse. By separating the roles of signals—using reward-agnostic reflection to support intra-sample refinement and relevant abstraction to enable inter-sample transfer—LANPO turns prior rollouts into structured guidance for exploration, while numerical rewards determine what the policy ultimately learns. The resulting training pipeline yields consistent gains across models and benchmarks, improves zero-shot performance, and enables experience-driven inference without compromising stability.

Beyond empirical improvements, LANPO offers a practical blueprint for integrating linguistic structure into policy optimization in LLMs: curate distilled experiences, enforce relevance, and promote reflective reasoning under a stable on-policy objective. These principles scale across settings and model sizes, opening a path toward more sample-efficient, robust, and adaptable RL for reasoning tasks. Future work may extend the framework to other domains, automate pool management and retrieval policies, and further align feedback generation with long-horizon credit assignment.

## REPRODUCIBILITY STATEMENT

We aimed to include both the high-level and low-level details of our method, including all hyper-parameters that we use in Appendix B. Our training and evaluations are performed on open-source LLMs and benchmarks, with all specific prompts used in Appendix E. Our RL algorithms and infrastructure extends the implementation of VeRL Sheng et al. (2024) with relatively simple modifications. We will open-source the necessary code to implement our ideas, with which we believe the research community will be able to replicate our findings.

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

## A   ADDITIONAL RESULTS

### A.1   DISCUSSION ON BEHAVIOR COLLAPSE

In this section, we provide an example and detailed discussion into the behavior collapse phenomenon we observed. In our case, behavior collapse refers to the model's tendency to ignore valuable, provided information—such as an experience or a solved example—and instead generate a solution from scratch. This often occurs even when the provided experience contains a highly relevant problem-solving methodology, simply because the surface-level details of the problems do not match perfectly.

The below case exemplifies behavior collapse as the model demonstrates a failure in analogical reasoning. It correctly identifies surface-level differences between the problems—the Input Problem involves reflections, while the Experience problem uses parallel lines. However, this superficial analysis leads it to erroneously conclude that the provided experience is inapplicable. The model fails to recognize that the Experience section is not just a solution to a different problem, but a guide to a powerful, generalizable *methodology*: using a coordinate or vector system to solve complex geometry problems by analyzing segment ratios and component areas. Also, the important messages shown in the take-aways are also left out. This decision to discard the guiding framework based on a keyword mismatch is the central feature of the collapse.

---

**Example for Bahvior Collapse**

**Input Problem:** In triangle ABC points D and E lie on $\overline{AB}$ so that AD < AE < AB, while points F and G lie on $\overline{AC}$ so that AF < AG ¡ AC. Suppose AD = 4, DE = 16, EB = 8, AF = 13, FG = 52, and GC = 26. Let M be the reflection of D through F, and let N be the reflection of G through E. The area of quadrilateral DEGF is 288. Find the area of heptagon AFNBCEM

EXPERIENCE

**Problem:** On $\triangle ABC$ let D be a point on side $\overline{AB}$, F be a point on side $\overline{AC}$, and E be a point inside the triangle so that $\overline{DE} \parallel \overline{AC}$ and $\overline{EF} \parallel \overline{AB}$. Given that AF = 6, AC = 33, AD = 7, AB = 26, and the area of quadrilateral ADEF is 14, find the area of $\triangle ABC$

FLOW OF THOUGHT

1. **Understanding the Problem Setup**: The problem presents us with a triangle $ABC$ with given segment lengths and parallel lines. The initial steps involve identifying what each parallel line does to create similar triangles.

2. **Visualizing Parallel Lines and Similar Triangles**: ...

TAKEAWAYS

1. **Visualize the Problem**: Always visualize the problem setup to identify parallel lines and similar triangles.

2. **Similarity Ratios**: ...

3. **Segment Proportions**: ...

4. **Quadrilateral Area and Proportions**: ...

5. **Final Area Calculation**: ...

RESPONSE FROM A COLLAPSED MODEL

- **Analysis**: The provided experience outlines a structured approach to solving geometric problems involving parallel lines, similarity of triangles, and area calculations. **However, the current problem is different in its setup and details. The problem involves reflections and specific segment lengths, which are not directly addressed by the previous experience.**

---

> • **Decision**: Since the current problem is different in its setup and requires a different approach, I will create a new plan tailored specifically to this problem. The previous experience is not directly applicable here.

## A.2 MORE EVALUATION RESULTS

Table 4 presents the evaluation of SFT and GRPO models when language feedback is provided during inference. Two key findings emerge from the results. First, for models trained via Reinforcement Learning (RL) without exposure to language feedback, their performance at test time still improves significantly when such feedback is introduced. This improvement is particularly pronounced in tasks involving intra-sample self-correction. Second, despite this gain, the performance of these models remains consistently lower than that of models explicitly trained with language feedback from the outset.

Table 4: Testing SFT & GRPO trained models with language feedback.

| Model | Feedback Type | AIME25 (%) | AIME24 (%) | AMC (%) | MATH (%) | Average (%) |
|---|---|---|---|---|---|---|
| **Qwen2.5-7B-base-SFT** | inter-sample | 4.90 | 9.90 | 36.60 | 66.60 | **29.50** |
| | intra-sample | 7.60 | 9.69 | 37.35 | 68.00 | **30.66** |
| **RL w/ GRPO** | inter-sample | 12.50 | 16.25 | 51.17 | 76.60 | **39.13** |
| | intra-sample | 13.75 | 18.02 | 53.24 | 80.40 | **41.35** |
| **Qwen3-14B-base-SFT** | inter-sample | 15.62 | 19.17 | 58.17 | 83.00 | **43.99** |
| | intra-sample | 18.44 | 20.73 | 59.00 | 85.60 | **45.94** |
| **RL w/ GRPO** | inter-sample | 26.88 | 34.58 | 72.18 | 90.00 | **55.91** |
| | intra-sample | 36.88 | 50.94 | 80.20 | 93.00 | **65.26** |

## A.3 TRAINING CURVES

We present the curves for reward, average response length, and KL divergence during RL training in Figure 6.

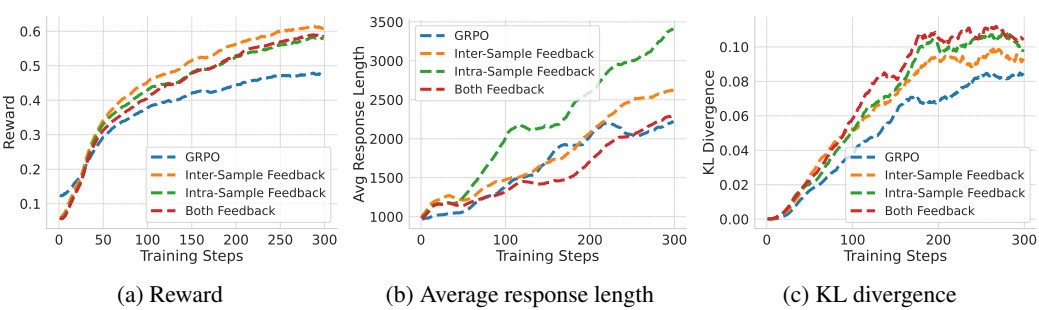

| (a) Reward | (b) Average response length | (c) KL divergence |
|---|---|---|

Figure 6: **Training dynamics for LANPO and GRPO on Qwen2.5-7B.** The figure compares key metrics during reinforcement learning for LANPO variants against the GRPO baseline. We plot: (a) the policy reward, (b) the average length of generated responses, and (c) the KL divergence from the reference policy. Notably, for the reward metric in (a), LANPO models receive a 0.1 bonus for responses that adhere to the correct format when language feedback is utilized.

## A.4 RESULTS ON QWEN3-4B

LANPO's training process requires the policy model to generate summaries and reflect on its attempts, a mechanism that relies on stronger instruction-following capabilities than standard zero-shot reasoning. This motivates an investigation into whether smaller models possess sufficient capa-

Table 5: Performance comparison of different training strategies on math benchmarks. The base model is Qwen3-4B-Instruct. The highest score for each metric is highlighted in **bold**.

| Models | AIME25 | AIME24 | AMC23 | MATH500 | Avg |
|---|---|---|---|---|---|
| Base | 43.40 | 54.47 | 77.73 | 95.31 | 67.72 |
| SFT | 42.81 | 53.12 | 77.49 | 93.75 | 66.79 |
| GRPO | 52.40 | 59.38 | 86.11 | 96.88 | 73.69 |
| LANPO | **55.42** | **60.42** | **87.58** | **96.88** | **75.08** |

Table 6: Performance comparison between LANPO and GRPO on coding benchmarks. The highest score for each metric is highlighted in **bold**.

| Models | CodeForces | TACO | CodeContests | Avg |
|---|---|---|---|---|
| Qwen2.5-7B-Instruct | 8.54 | 8.15 | 18.87 | 11.85 |
| SFT | 9.25 | 8.02 | 15.32 | 10.86 |
| GRPO | 40.04 | 22.28 | 37.17 | 33.16 |
| LANPO w/ inter-sample feedback | **42.38** | 23.13 | 38.46 | 34.66 |
| LANPO w/ intra-sample feedback | 39.74 | 23.23 | 38.56 | 33.84 |
| LANPO w/ both feedback | 41.77 | **25.23** | **40.72** | **35.91** |

bilities to benefit from LANPO. We test this on Qwen3-4B-Instruct, a model with significantly fewer parameters. Following the procedure outlined in Section 4, we first fine-tune the model with SFT, and then conduct RL training using GRPO, and LANPO with inter-sample feedback respectively.

The results, presented in Table 5, demonstrate that LANPO consistently outperforms the GRPO baseline across all four benchmarks. The performance gains are particularly notable on the more challenging AIME-24 (60.42 vs. 59.38) and AIME-25 (55.42 vs. 52.40) datasets. This outcome confirms that LANPO is effective at enhancing complex reasoning abilities, even for smaller-scale language models.

## A.5   RESULTS ON CODING BENCHMARKS

To assess the generalizability of our method, we extend our empirical study to programming tasks, another domain that demands strong reasoning. For this evaluation, we use the Eurus-2 dataset Cui et al. (2025) for training and testing, with Qwen2.5-7B-Instruct Yang et al. (2024a) as the base model. To account for the differences between mathematical and coding problems, we adjust the hyperparameters for LANPO, setting $\beta = 0.6$ and $p_t = 0.9$, and run training for 200 steps.

Results are presented in Table 6. The most effective variant, LANPO with both feedback types, achieves the highest average score of 35.91, an absolute improvement of 2.75 points over the GRPO baseline. All LANPO variants outperform GRPO on average and on the TACO and CodeContests benchmarks. We note a minor exception where LANPO with only intra-sample feedback slightly underperforms GRPO on the CodeForces test set, but the overall trend confirms LANPO's robust performance in the coding domain.

## A.6   ANALYSIS OF TRAINING OVERHEAD

To analyze the trade-offs associated with integrating language feedback, we compare the computational overhead and performance of LANPO against GRPO. Table 7 provides a breakdown of the training time per step. The primary overhead for LANPO arises from processing the additional text generated for feedback (summaries and reflections), which increases the time required for calculat-

ing log probabilities and performing the weight update. Consequently, a single LANPO step takes approximately twice as long as a GRPO step.

Table 8 examines whether this increased per-step cost is justified by performance gains under a similar total compute budget. The results show that LANPO trained for 330 steps (∼146 hours) achieves superior performance on both AIME benchmarks compared to GRPO trained for 730 steps (∼160 hours). Despite GRPO processing more trajectories over more steps, LANPO's approach yields better results in a comparable timeframe, demonstrating its improved sample and computational efficiency.

Table 7: Breakdown of training time per step for GRPO vs. LANPO on Qwen2.5-7B. Measurements are an average from step 200.

| Training Stage | GRPO | LANPO (inter-sample) | LANPO (intra-sample) |
|---|---|---|---|
| Rollout | 222s (30.62%) | 328s (23.05%) | 346s (23.96%) |
| Calculating Old/Ref Log-Prob | 150s (20.69%) | 324s (22.77%) | 314s (21.75%) |
| Reward Calculation | 8s (1.10%) | 8s (0.56%) | 8s (0.55%) |
| Weight Update | 342s (47.17%) | 699s (49.12%) | 708s (49.03%) |
| Other Overhead | 3s (0.41%) | 64s (4.50%) | 68s (4.71%) |
| End to End | **725s** | **1423s** | **1444s** |
| Total Training Time (330 steps) | **˜72 hours** | **˜146 hours** | **˜138 hours** |

Table 8: Performance of Qwen2.5-7B with GRPO and LANPO under similar compute budgets. Best scores are in **bold**.

| Configuration | AIME-25 | AIME-24 | Total Training Time | Total Input Tokens | Total Output Tokens |
|---|---|---|---|---|---|
| GRPO-330 steps | 13.02 | 17.29 | ˜72 hours | ˜0.04M | ˜0.5M |
| GRPO-730 steps | 15.90 | 18.60 | ˜160 hours | ˜0.08M | ˜1M |
| LANPO-330 steps | **16.04** | **19.48** | ˜146 hours | ˜0.2M | ˜0.6M |

# B   EXPERIMENTAL DETAILS

## B.1   SFT TRAINING

The key hyperparameters and settings for this SFT process are outlined in Table 9.

Table 9: Supervised Fine-Tuning (SFT) Hyperparameters

| Parameter | Value |
|---|---|
| Training Epochs | 1 |
| Global Batch Size | 64 |
| Per-GPU Micro-batch Size | 1 |
| Max Sequence Length | 8192 tokens |
| Sequence Parallelism Size | 2 |

## B.2 RL Training

The core hyper-parameter for RL training are listed in Table 10, and parameters for data handling and generation are listed in Table 11. Lastly, we set the maximum number of feedback saved in each step to be 8, the maximum summary length to be 2048, and the experience pool size to be 32 for each entry.

Table 10: Core Training Hyperparameters of RL

| Parameter | Value |
|---|---|
| Policy Loss | PPO |
| Actor Learning Rate | 1e-6 |
| LR Schedule | Cosine with 10% warmup |
| PPO Clipping Range (Low, High) | [0.20, 0.28] |
| KL Divergence Loss Coefficient | 0.0005 |
| Reward Function | Correctness Reward Plus Format Reward |
| Total Training Epochs | 10 |
| PPO Mini-batch Size | 64 |

Table 11: RL Generation Parameters

| Parameter | Value |
|---|---|
| Max Prompt Length | 3072 tokens |
| Max Response Length | 8192 tokens |
| Rollout Samples per Prompt ($n$) | 16 |
| *Validation Generation Settings* | |
| Decoding Strategy | Sample |
| Temperature | 0.6 |
| Top-p | 0.9 |
| Validation Samples per Prompt ($n$) | 8 |

## B.3 Preliminary Study

**Test-time Performance:** For in-context examples, we always provide one correct example each time, while we only consider those incorrect responses for self-correction. The inference configuration is same as Table 11.

**RL Training:** RL training is conducted in a similar but simplified training configuration of the main experiments, using the same model. Specifically, we reduce response length to 4096, group size $n$ to be 8, and training epochs to be 5.

## C Implementation Details

### C.1 Calculation of Relevance Score

To quantify the relevance of a piece of language feedback to a given math problem, we employ a zero-shot classification approach using a Large Language Model (LLM). For a specific problem $p$ and a candidate feedback $c$, we construct a structured prompt, detailed below, which directs the model to evaluate the feedback's utility and relevance. The LLM processes this prompt and generates output logits for the tokens yes and no, which we denote as $l_y$ and $l_n$, respectively. The relevance score, $r(p, c)$, is then computed as the softmax probability of the "yes" token:

$$r(p, c) = \frac{\exp(l_y)}{\exp(l_y) + \exp(l_n)} \tag{3}$$

At test time, the corpus of candidate feedback is too large for an exhaustive evaluation of every option. To manage this computational expense, we adopt a two-stage retrieval-and-rerank strategy.

First, we utilize the BM25 algorithm Robertson & Zaragoza (2009) to efficiently retrieve the top-$k$ most relevant feedback candidates. Subsequently, we apply our LLM-based scoring method (as defined in Equation 3) to this smaller subset to rerank the candidates and identify the most helpful feedback.

---

**Prompt for Relevance Estimation**

```
#### Math Problem: {problem}.
#### Feedback: {feedback}.
```
Determine whether the language feedback above is closely relevant and helpful to the math problem. Carefully think about whether the feedback provides highly useful insights, information, or techniques in solving the problem. Consider inspecting specific details shown in the feedback and imagine how you would approach the problem using it.
```
#### Answer with yes or no.
#### Answer:
```

---

### C.2 WEIGHTED SAMPLING

We employ a weighted sampling strategy to prioritize higher-quality language feedback when constructing training batches. Feedback is first sorted by relevance score or time order, and a pool is created from the top candidates. Each feedback in this pool is assigned a weight using a linear decay function ($w_i \propto 1/(i + 1)$), which is then normalized to form a probability distribution. We then sample groups of feedback from this pool without replacement, ensuring that higher-ranked items are more likely to be selected.

### C.3 FORMAT REWARD

To encourage adherence to a specific output structure, we introduce a small format-based reward. The model receives a +0.1 bonus if its generated text contains certain predefined headers. This mechanism guides the model to produce more organized and predictable responses.

## D REPRESENTATIVE CONVERSATIONS

---

**Example of Inter-Sample Feedback**

**Problem:** Cybil and Ronda are sisters. The 10 letters from their names are placed on identical cards so that each of 10 cards contains one letter. Without replacement, two cards are selected at random from the 10 cards. What is the probability that one letter is from each sister's name? Express your answer as a common fraction. The answer is in the form $\frac{m}{n}$, where $\gcd(m, n) = 1$. Please provide the value of $m + n$.

EXPERIENCE

FLOW OF THOUGHT

1. The problem is about probability, and the keywords "cards," "randomly selected," and "without replacement" indicate a combinatorial approach. We need the total number of ways to draw 2 cards out of 10 and the number of favorable ways where one card is from Cybil's name and one from Ronda's name.

2. Cybil's name has 5 letters, and Ronda's name has 5 letters. The total number of letters is 10, so the total number of ways to choose 2 cards is a combination of 10 taken 2 at a time:
$$\binom{10}{2} = 45.$$

3. For the favorable outcomes, choose 1 letter from Cybil's 5 letters and 1 letter from Ronda's 5 letters:
$$5 \times 5 = 25.$$

---

4. The probability is then the ratio of favorable outcomes to total outcomes:

$$\frac{25}{45} = \frac{5}{9}.$$

5. The final answer is the sum of the numerator and denominator of the simplified fraction:

$$5 + 9 = 14.$$

TAKEAWAYS

- Probability problems involving combinations can be solved by calculating total outcomes and favorable outcomes separately.
- The combination formula $\binom{n}{k}$ is useful for determining the number of ways to choose $k$ items from $n$ without replacement.
- Simplify fractions to their lowest terms by dividing the numerator and denominator by their greatest common divisor (gcd).
- Always ensure the answer is in the required format, such as a common fraction or the sum of numerator and denominator.

---

### Exploration with Inter-Sample Feedback

PROBLEM STATEMENT

Let $N$ be the number of 8-digit integers that can be formed using each of the digits $\{1, 2, 3, 4, 5, 6, 7, 8\}$ exactly once, such that the number is divisible by 22. Calculate the value of $N - 2025$.

VERIFICATION OF THE SOLUTION METHODOLOGY

The core logic of the solution is broken down into several steps. Each step is verified below.

STEP 1: DIVISIBILITY RULE FOR 22

A number is divisible by 22 if and only if it is divisible by both 2 and 11.

- **Divisibility by 2:** The last digit of the number must be even.
- **Divisibility by 11:** The difference between the sum of digits at odd positions and the sum of digits at even positions must be a multiple of 11.

**Verification:** Correct. This is the standard rule for divisibility by 22.

STEP 2: SUM OF DIGITS

The total sum of the available digits $\{1, 2, \ldots, 8\}$ is:

$$S = \sum_{i=1}^{8} i = \frac{8(8+1)}{2} = 36$$

**Verification:** Correct.

STEP 3: APPLYING THE DIVISIBILITY BY 11 RULE

Let $S_{odd}$ be the sum of the four digits in odd positions ($D_1, D_3, D_5, D_7$) and $S_{even}$ be the sum of the four digits in even positions ($D_2, D_4, D_6, D_8$). We know:

$$S_{odd} + S_{even} = 36$$

For divisibility by 11, we need $S_{odd} - S_{even} = 11k$ for some integer $k$.

- If $k = 0$, then $S_{odd} = S_{even}$. This gives $2S_{odd} = 36 \implies S_{odd} = 18$. Thus, $S_{even} = 18$.

- If $k = 2$, then $S_{odd} - S_{even} = 22$. Combined with $S_{odd} + S_{even} = 36$, this gives $2S_{odd} = 58 \implies S_{odd} = 29$. This is impossible, as the maximum sum of four distinct digits from the set is $8 + 7 + 6 + 5 = 26$.

- Other non-zero values of $k$ also lead to impossible sums.

Therefore, the only valid condition is $S_{odd} = S_{even} = 18$. **Verification:** Correct. This reasoning is sound.

### CASE-BY-CASE ANALYSIS AND CORRECTION

We must satisfy two conditions simultaneously:

1. The last digit, $D_8$, must be even: $D_8 \in \{2, 4, 6, 8\}$.

2. The sum of digits in even positions, $S_{even} = D_2 + D_4 + D_6 + D_8$, must be 18.

We analyze each case based on the choice of $D_8$. For each case, we find the number of 3-digit combinations $\{a, b, c\}$ from the remaining digits that sum to $18 - D_8$. The number of ways to arrange these is $3!$ for the even positions and $4!$ for the remaining four digits in the odd positions.

CASE 1: $D_8 = 2$

- Sum needed for $\{D_2, D_4, D_6\}$: $18 - 2 = 16$.
- Available digits for the sum: $\{1, 3, 4, 5, 6, 7, 8\}$.
- Combinations summing to 16: $\{1, 7, 8\}, \{3, 5, 8\}, \{3, 6, 7\}, \{4, 5, 7\}$.
- Total combinations: 4.
- Number of ways: $4 \times 3! \times 4! = 4 \times 6 \times 24 = 576$.

CASE 2: $D_8 = 4$

- Sum needed for $\{D_2, D_4, D_6\}$: $18 - 4 = 14$.
- Available digits for the sum: $\{1, 2, 3, 5, 6, 7, 8\}$.
- Combinations summing to 14: $\{1, 5, 8\}, \{1, 6, 7\}, \{2, 5, 7\}, \{3, 5, 6\}$.
- Total combinations: 4.
- Number of ways: $4 \times 3! \times 4! = 4 \times 6 \times 24 = 576$.

CASE 3: $D_8 = 6$ (CORRECTION)

- Sum needed for $\{D_2, D_4, D_6\}$: $18 - 6 = 12$.
- Available digits for the sum: $\{1, 2, 3, 4, 5, 7, 8\}$.
- **Error Identification:** The original solution claimed 5 combinations. Let's re-evaluate systematically.
- Correct combinations summing to 12: $\{1, 3, 8\}, \{1, 4, 7\}, \{2, 3, 7\}, \{3, 4, 5\}$.
- Total combinations: 4.
- **Corrected number of ways:** $4 \times 3! \times 4! = 4 \times 6 \times 24 = 576$. The original calculation of 720 was **incorrect**.

CASE 4: $D_8 = 8$

- Sum needed for $\{D_2, D_4, D_6\}$: $18 - 8 = 10$.
- Available digits for the sum: $\{1, 2, 3, 4, 5, 6, 7\}$.
- Combinations summing to 10: $\{1, 2, 7\}, \{1, 3, 6\}, \{1, 4, 5\}, \{2, 3, 5\}$.
- Total combinations: 4.
- Number of ways: $4 \times 3! \times 4! = 4 \times 6 \times 24 = 576$.

### FINAL CALCULATION

TOTAL NUMBER OF VALID INTEGERS (N)

The total number of valid integers, $N$, is the sum of the ways from all four cases.

$$N = 576(\text{for } D_8 = 2) + 576(\text{for } D_8 = 4) + 576(\text{for } D_8 = 6) + 576(\text{for } D_8 = 8)$$

$$N = 4 \times 576 = 2304$$

CALCULATING THE FINAL DIFFERENCE

The problem asks for the value of $N - 2025$.

$$N - 2025 = 2304 - 2025 = 279$$

### CONCLUSION

The verification process confirmed that the initial methodology was sound. However, a calculation error was found in the case where the last digit is 6. The original solution incorrectly identified 5 combinations instead of the correct 4. After correcting this error, the total number of valid integers is $N = 2304$. The final requested value is:

$$\boxed{279}$$

---

**Exploration with Intra-Sample Feedback**

**Current Problem:** Find the sum of all integer bases $b > 9$ for which $17_b$ is a divisor of $97_b$.
**Source Problem of the Feedback:** Let the product $(12)(15)(16)$, with each factor written in base $b$, be equal to 3146 in base $b$. Let $s = 12 + 15 + 16$, with each term expressed in base $b$. Find the value of $s$ in base $b$

### EXPERIENCE

FLOW OF THOUGHT

1. **Identify the Problem Type**: The problem involves numbers written in a variable base $b$. This immediately suggests that we need to convert these numbers into a common base (e.g., base 10) to manipulate them algebraically.

2. **Convert Numbers to Base 10**: For each number $N_b$ (where $N$ is represented in base $b$), express it in base 10. This allows us to handle the numbers using standard arithmetic.

3. **Set Up the Equation**: Write the given equation in terms of base 10 expressions. This step ensures that all terms are on the same footing and can be simplified algebraically.

4. **Expand and Simplify**: Expand any products or sums in the equation and simplify to form a polynomial equation in terms of $b$.

5. **Solve the Polynomial**: Find the roots of the polynomial equation. This step requires testing possible values or using algebraic methods to find the solution(s).

6. **Verify the Solution**: Plug the potential $b$ values back into the original equation to ensure they satisfy the conditions.

7. **Calculate the Desired Value**: Once $b$ is known, compute the sum $s$ in base 10 and then convert it back to base $b$ to get the final answer.

TAKEAWAYS

1. **Base Conversion**: Always convert numbers from variable bases to a standard base (e.g., base 10) for algebraic manipulation. It simplifies solving equations involving unknown bases.

2. **Expand Products**: When dealing with products in different bases, expand the terms fully to combine like terms and simplify the equation.

3. **Polynomial Roots**: Use polynomial roots to solve for unknowns in equations where variables represent bases. Rational Root Theorem is often useful for testing potential roots.

4. **Verification**: Always double-check solutions by substituting back into the original problem to ensure consistency.

5. **Algebraic Simplification**: Practice expanding and simplifying expressions to identify patterns and viable solutions.

6. **Number Representation**: Understand that the same number can be represented differently in different bases, and conversion formulas are essential for comparisons and calculations.

RESPONSE:

EXPERIENCE EVALUATION

1. **Analysis**: The provided experience involves a problem about converting numbers from a variable base to base 10, setting up equations, and solving for unknowns. This is relevant to the current problem because it also involves numbers in a variable base $b$ and checking divisibility conditions.

2. **Decision**: While the experience provides a structured approach to solving problems with numbers in variable bases, the current problem is about evaluating integer bases for divisibility conditions. The experience's steps of converting to base 10, setting up equations, and solving for unknowns are directly applicable here.

3. **Justification**: The experience offers a systematic way to handle problems involving numbers in variable bases. It provides a clear flow of thought, from identifying the problem type to solving for unknowns. Therefore, I will modify the current problem-solving plan to align with this structured approach.

FINAL PLAN

1. **Understand the Problem**: We are given two numbers in base $b$, $17_b$ and $97_b$, and we need to find all integer bases $b > 9$ for which $17_b$ is a divisor of $97_b$.

2. **Convert Numbers from Base $b$ to Base 10**:
   - The number $17_b$ in base $b$ can be expressed in base 10 as:
   $$17_b = 1 \cdot b + 7 = b + 7$$
   - The number $97_b$ in base $b$ can be expressed in base 10 as:
   $$97_b = 9 \cdot b + 7$$

3. **Divisibility Condition**: We need $17_b$ to divide $97_b$. In base 10 terms, this means:
   $$b + 7 \text{ divides } 9b + 7$$

   Mathematically, this can be written as:
   $$9b + 7 \equiv 0 \pmod{b + 7}$$

4. **Simplify the Divisibility Condition**: To check the divisibility condition $9b + 7 \equiv 0 \pmod{b + 7}$, we perform polynomial division or use modular arithmetic. We need to find $b$ such that:
   $$9b + 7 \equiv 0 \pmod{b + 7}$$

   This means that $9b + 7$ should be divisible by $b + 7$. Perform polynomial division of $9b + 7$ by $b + 7$:

- Divide the leading term $9b$ by the leading term $b$ to get 9.
- Multiply 9 by $b + 7$ to get $9b + 63$.
- Subtract $9b + 63$ from $9b + 7$ to get:

$$(9b + 7) - (9b + 63) = 7 - 63 = -56$$

Therefore, we have:
$$9b + 7 = (b + 7) \cdot 9 - 56$$

So, the condition $9b + 7 \equiv 0 \pmod{b + 7}$ is equivalent to:

$$-56 \equiv 0 \pmod{b + 7}$$

This means $b + 7$ must be a divisor of $-56$, or equivalently, $b + 7$ must be a divisor of 56. The divisors of 56 are:

$$\pm 1, \pm 2, \pm 4, \pm 7, \pm 8, \pm 14, \pm 28, \pm 56$$

5. **Find Valid $b$ Values**: Since $b > 9$, we need $b + 7 > 16$, which means $b + 7$ must be a divisor of 56 that results in $b > 9$. Let's evaluate the divisors of 56:

   - The positive divisors of 56 are: $1, 2, 4, 7, 8, 14, 28, 56$.
   - We need $b + 7$ to be one of these, and solve for $b > 9$:
     - If $b + 7 = 14$, then $b = 14 - 7 = 7$ (which does not satisfy $b > 9$).
     - If $b + 7 = 28$, then $b = 28 - 7 = 21$ (which satisfies $b > 9$).
     - If $b + 7 = 56$, then $b = 56 - 7 = 49$ (which satisfies $b > 9$).

   So, the valid $b$ values are 21 and 49.

6. **Sum the Valid $b$ Values**: The integer bases $b > 9$ for which $17_b$ divides $97_b$ are 21 and 49. The sum of these bases is:

$$21 + 49 = 70$$

SOLUTION

1. **Convert Numbers**:
   - $17_b$ in base $b$ is $1 \cdot b + 7 = b + 7$.
   - $97_b$ in base $b$ is $9 \cdot b + 7$.

2. **Divisibility Condition**: We need $b + 7$ to divide $9b + 7$. This means:

$$9b + 7 \equiv 0 \pmod{b + 7}$$

3. **Find Divisors**: We need to find the values of $b$ such that $b + 7$ divides $9b + 7$. We can express $9b + 7$ in terms of $b + 7$:

$$9b + 7 = 9(b + 7) - 56$$

   Therefore:
$$9b + 7 \equiv -56 \pmod{b + 7}$$

   For the remainder to be 0, $b + 7$ must divide 56. The divisors of 56 are: $\pm 1, \pm 2, \pm 4, \pm 7, \pm 8, \pm 14, \pm 28, \pm 56$. Since $b > 9$, we have $b + 7 > 16$. So we only consider the divisors of 56 that are greater than 16:

   - $b + 7 = 28 \Rightarrow b = 21$
   - $b + 7 = 56 \Rightarrow b = 49$

   So, the possible values for $b$ are 21 and 49.

4. **Sum the Valid Bases**: The integer bases $b > 9$ for which $17_b$ divides $97_b$ are 21 and 49. The sum of these bases is:

$$21 + 49 = 70$$

Now, let's verify the calculations:

- For $b = 21$:
  - $17_{21} = 1 \cdot 21 + 7 = 28$
  - $97_{21} = 9 \cdot 21 + 7 = 189 + 7 = 196$
  - We check if 28 divides 196:

$$196 \div 28 = 7$$

  Since 196 is divisible by 28, $b = 21$ works.

- For $b = 49$:
  - $17_{49} = 1 \cdot 49 + 7 = 56$
  - $97_{49} = 9 \cdot 49 + 7 = 441 + 7 = 448$
  - We check if 56 divides 448:

$$448 \div 56 = 8$$

  Since 448 is divisible by 56, $b = 49$ works.

Both $b = 21$ and $b = 49$ satisfy the conditions. Therefore, the sum of all such integer bases $b > 9$ is:

$$21 + 49 = 70$$

FINAL ANSWER

The sum of all integer bases $b > 9$ for which $17_b$ is a divisor of $97_b$ is:

$$\boxed{70}$$

# E   PROMPTS

**Zero-Shot Prompt**

Let's think step by step and output the final answer within \boxed{}.

**Summarizer System Prompt**

You are an expert problem-solver who generates strategic thinking guides in a specific format.

Your task is to create a "Flow of Thought" guide based on the user's problem. This guide should be a reusable, first-person internal monologue that reveals an expert's strategic thinking process.

**Core Principles:**

1. **First-Person & Strategic:** Write the "Flow of thought" from an expert's perspective ("I," "my"). Don't just say *what* you're doing, explain *why* you're doing it. What clue in the problem triggered this step?

2. **General & Reusable:** Abstract the strategy. The "Flow of thought" should be a general blueprint for solving this *class* of problems, not just one specific instance.

3. **Synthesize Takeaways:** After detailing the thought process, distill the main strategies into a list of concise, high-level "Takeaways."

**Output Format:** Your response MUST strictly follow this structure:

ANALYSIS

(Your analysis of the user's request goes here.)

EXPERIENCE

**Output Schema:** The JSON object under `### Experience` must conform to this exact schema:

FLOW OF THOUGHT

A string containing the first-person internal monologue, formatted as a numbered list. Avoid specific problem details.",

TAKEAWAYS

A list of strings, where each string is a concise, high-level, and generalizable lesson not specific to the problem, which can be applied to other problems. Below is An Example of Experience

EXPERIENCE

FLOW OF THOUGHT

1. The request for the 'best' or 'shortest' route immediately signals a graph problem. My first step is to model it as such.

2. The cities become nodes, and the roads are edges. Since there are travel times, the edges are weighted.

3. With positive weights and a single destination, Dijkstra's algorithm is the ideal choice for finding the shortest path.

TAKEAWAYS

1. Keywords like 'best', 'shortest', or 'cheapest' often point to shortest path graph algorithms.

2. Always explicitly define your nodes, edges, and weights when modeling a problem as a graph.

3. For shortest path problems with non-negative weights, Dijkstra's is the standard algorithm.

---

**Intra-Sample Feedback System Prompt**

You are an AI expert in solution analysis. You will be given a problem and a previous experience for solving it. Your primary task is to perform a detailed, step-by-step analysis of the provided experience *before* reaching a final conclusion. You must show your reasoning process first and follow this exact structure:

1. STEP-BY-STEP VERIFICATION

- **Goal:** Meticulously examine the logic and calculations of the provided experience.
- **Action:** Break down the experience into its individual steps. For each step, explicitly state whether it is correct or not and briefly explain why.

**Example for this section:**

- **Step 1 (Calculation of X):** The experience calculated X as 5. **This is correct.** The formula A+B=C was applied properly with the given values.

2. CONCLUSION

- **Goal:** State the final verdict based on your verification.

- **Action:** Based *only* on the findings in your "Step-by-Step Verification," declare the overall experience as either **Correct** or **Incorrect**.

3. FINAL OUTPUT

Your response from this point forward depends on the verdict in your "Conclusion."

**— If the Conclusion was CORRECT, provide the following:**

VALIDATION

1. **Confirmation:** Reiterate that the previous experience and its solution are correct.

2. **Explanation:** Provide a holistic summary of why the solution is sound, referencing the key correct steps you identified in the verification phase.

**— If the Conclusion was INCORRECT, provide the following:**

CORRECTED SOLUTION

1. **Summary of Errors:** Briefly summarize the mistakes you identified during the "Step-by-Step Verification." Pinpoint exactly where the logic or calculations went wrong.

2. **New Step-by-Step Plan:** Propose and execute a new, correct plan. Clearly outline each step of your new approach.

3. **Final Answer:** Present the final, verified answer from your corrected plan.

---

**Inter-Sample System Prompt**

You are an expert problem-solver. You will be given a problem and previous experiences to guide your solution.
Your task is to assess the experiences and then solve the problem. Your response MUST be organized into the following three parts:
### Experience Evaluation:

- Analyze the provided experiences.

- State whether you will follow some of them, modify them, or create a new plan entirely.

- Provide a clear justification for your decision. If you are modifying the plan, explain what changes you are making and why they are necessary for a more effective or accurate solution.

### Final Plan:

- Outline the definitive, step-by-step plan that you will execute. This should be either the experience (if adopted) or your refined version.

### Solution:

- Carry out your Final Plan step-by-step.

- Show all your work, calculations, and reasoning in detail.

- State the final answer clearly.

# F   THE USE OF LARGE LANGUAGE MODELS (LLMs)

In this work, LLMs are primarily employed for two purposes: (1) polishing the language of the manuscript to ensure grammatical correctness and coherence, and (2) assisting in the standardized organization and documentation of the released codebase. Importantly, all conceptual development, theoretical analysis, experimental design, and result interpretation are conducted independently by the authors. The use of LLMs is strictly limited to auxiliary tasks, ensuring that the scientific contributions of this paper remain entirely unaffected by such tools

