# OpenReview forum: "Bootstrapping Language and Numerical Feedback for Reinforcement Learning in LLMs"
_ICLR.cc/2026/Conference — Submitted to ICLR 2026_

### Official Review · Reviewer_Wmpd · 2025-10-28

**Soundness:** 3
**Presentation:** 2
**Contribution:** 4
**Rating:** 6
**Confidence:** 4

**Summary:**

This paper proposes a method to integrate natural language feedbacks in on-policy RL training. Specifically, this method introduces two methods: reward-agnostic reflection and relevant abstraction. The former serves to do self-reflection on its previous roll-outs (without reward feedback) and facilitate more exploration; the latter summarizes policies from similar tasks and shrinks exploration space. On mathematical reasoning tasks, this work outperforms GRPO based methods.

**Strengths:**

1, Combining language and scalar feedbacks is a very interesting direction. And this work proposes an reasonable solution, which could be an important contribution to the research community.

2, I especially appreciate the empirical exploration on challenges in introducing language feedback (section 3). These experiments provide valuable insights on how pure language feedback affects training and test performance.

3, In section 5, empirical results show that this method could effectively improve vanilla GRPO performance by introducing language feedbacks.

**Weaknesses:**

1, My biggest concern is with the writing in Section 4. While the proposed framework is sound in general, there are unclear points after reading section 4. (I will list my questions in the question section.) I would recommend the authors to describe their method using more mathematical notations.

2, As pointed in the discussion section, the authors also admit that long context would be induced during training time, which will significantly increase training time. Considering that RL training is already time-consuming, further training time increase could be intimidating. Could you compare the wall-clock training time of vanilla GRPO and your time (using the same hardware)?

3, The last point is less concerning, but worth pointing out that the reflection and summarization actually requires LLMs with decent language capabilities. So, the proposed method might not be effective for small size LMs. Could you run an experiments with Qwen3-4B model?

Minor:
1, The “strong ICL benefit” in Figure 2 caption is unclear by reading the figure. Please consider using “inter-sample feedback”, or other appropriate terms.

**Questions:**

1, The first question is that whether the reflection and the contexts will be used to compute the gradient and back-propagated?

2, For the self-reflection process, what are the input and outputs? What languages/tokens are added to the RL training context? Will there be a distribution shift if no intra-sample reflection is conducted during inference time?

3, Details in Section 4:
- in line 261, What is the “feedback”?
- in line 263, what is the “high-level solution plan”?
- in line 283, what do you mean “preserving independence”?
- in line 286 to line 288, the loss/negative reward is significantly different from that of GRPO. Can you explain why you make those changes?

---

> ### Author Response · Authors · 2025-11-23
> **Rebuttal Responses to Reviewer Wmpd (1/3)**
>
> Dear Reviewer Wmpd
>
> We would like to first appreciate you for acknowledging **the significance of our empirical exploration** **and performance** of our proposed method, LANPO.  Following your insightful comments on our writing and the computational overhead of LANPO, we have prepared additional results and discussions below, with which we hope can address your concerns. Should you have any further questions, please feel free to raise them; all your feedback will help us improve our work.
>
> ---
>
> > **Q1:** Writings and notations in Section 4.
> >
>
> **A1: Clarification of the terms.** Sorry for the confusion. We would like to further clarify the terms that you mentioned, and have revised our manuscript accordingly.
>
> > **Q1.1:** in line 261, What is the “feedback”
> >
>
> **A1.1:** The “feedback” refers to experience retrieved from the experience pool, which are summarized from correct solutions to relevant problems.
>
> > **Q1.2:** in line 263, what is the “high-level solution plan”?
> >
>
> **A1.2:** The “high-level solution plan” is part of the model responses when utilizing inter-sample feedback (examples can be found in Appendix E). Specifically, we instruct the model to first generate a high-level plan with the inter-sample feedback, and then conduct the plan step-by-step within one context window.
>
> > **Q1.3:** in line 283, what do you mean “preserving independence”?
> >
>
> **A1.3:** “Preserving independence” means we try to preserve the model’s ability to reason independently, i.e., without any language feedback, by randomly setting the language feedback as empty during training.
>
> > **Q1.4:** in line 286 to line 288, the loss/negative reward is significantly different from that of GRPO. Can you explain why you make those changes?
> >
>
> **A1.4:** We apologize for the lack of clarity in our initial description of the objective function. To clarify, **LANPO does not modify the core GRPO objective**. The objective in our submitted manuscript is a simplified version of GRPO to stress the explicit incorporation of context $c$ during forward/backward propagation. The group-wise advantage and probability-ratio clipping are actually untouched. The key take-aways is that LANPO is an orthogonal enhancement, operating at a higher level by accumulating and utilizing the language feedback during RL, and is compatible with various PPO-style objectives (PPO, GRPO, etc.). Thanks for the question, and we have rewritten the objective accordingly.
>
> **Paper Update:** Thanks for your careful reading again, we have revised Section 4 to improve clearness of our terms and notations. Specifically, we have included more details in places pointed by Q1.1-Q1.3, and have rewritten the LANPO objective following Q1.4 with extra explanations for our modification to the original GRPO objective.

---

> ### Author Response · Authors · 2025-11-23
> **Rebuttal Responses to Reviewer Wmpd (2/3)**
>
> > **Q2:** As pointed in the discussion section, the authors also admit that long context would be induced during training time, which will significantly increase training time. Considering that RL training is already time-consuming, further training time increase could be intimidating. Could you compare the wall-clock training time of vanilla GRPO and your time (using the same hardware)?
> >
>
> **A2: Breakdown of Training Step Time:** The primary overhead of LANPO during training comes from processing longer input sequences when language feedback is included. We profiled a training step for GRPO vs. LANPO on a Qwen2.5-7B model, and the results are below:
>
> | Training Stage | GRPO | LANPO (inter-sample) | LANPO (intra-sample) |
> | --- | --- | --- | --- |
> | **Rollout** | 222s (30.62%) | 328s (23.05%) | 346s (23.96%) |
> | **Calculating Old/Ref Log-Prob** | 150s (20.69%) | 324s (22.77%) | 314s (21.75%) |
> | **Reward Calculation** | 8s (1.10%) | 8s (0.56%) | 8s (0.55%) |
> | **Weight Update** | 342s (47.17%) | 699s (49.12%) | 708s (49.03%) |
> | **Other Overhead (communication, experience pool update etc.)** | 3s (0.41%) | 64s (4.50%) | 68s (4.71%) |
> | **End to End** | **725s** | **1423s** | **1444s** |
> | **Total Training Time** | **~72 hours** | **~146 hours** | **~138 hours** |
>
> **Performance vs. Compute Budget:** A critical question is whether LANPO's gains are simply due to this increased computation. To investigate this, we trained the GRPO baseline for more steps (730 vs. 330) to closely match the total training time of LANPO (~160 hours vs. ~146 hours).
>
> | Qwen2.5 7B | AIME25 | AIME24 | Total Training Time | Total Input Tokens | Total Output Tokens |
> | --- | --- | --- | --- | --- | --- |
> | GRPO-330 steps | 13.02 | 17.29 | ~72 hours | ~0.04M | ~0.5M |
> | GRPO-730 steps | 15.9 | 18.6 | ~160 hours | ~0.08M | ~1M |
> | LANPO-330 steps (w/ Inter-sample) | **16.04** | **19.48** | ~146 hours | ~0.2M | ~0.6M |
>
> The results demonstrate that **LANPO outperforms the compute-matched GRPO baseline, especially on the more difficult AIME24 benchmark.** This, along with the training curves in Figure 5(a) showing LANPO's faster reward increase in terms of training steps, confirms that LANPO's performance gains stem from improved sample efficiency and more effective learning, not just from consuming more FLOPs.
>
> **Future Work on Acceleration:** In this work, we mainly aim to study whether incorporating language feedback into numerical RL has the potential to outperform the classic trial-and-error loop. Thus, we focus more on the comparison in terms of sample complexity, i.e., how many state-action-reward triplets are used to learn the policy, rather than the empirical time cost. We agree wall-clock efficiency is a critical aspect for the practical application of RL methods. For exciting avenues of optimization, one can experiment with condensing the experiences into shorter formats, or adopting shorter system prompts, which could further reduce the computational overhead and make LANPO even more practical.
>
> **Paper Update:** Following your question,  we have incorporated concrete discussions and results about the computational efficiency of LANPO into the manuscript as Table 7 and Table 8 in Appendix A.
>
> ---
>
> > **Q3:** The last point is less concerning, but worth pointing out that the reflection and summarization actually requires LLMs with decent language capabilities. So, the proposed method might not be effective for small size LMs. Could you run an experiments with Qwen3-4B model?
> >
>
> **A3: New Results on a Smaller Model (Qwen3-4B).** To test the effectiveness of LANPO on relatively small LLMs, we conduct additional experiments with Qwen3-4B-Instruct-2507 on the math reasoning tasks. Due to the tight time constraint of the rebuttal period, we are currently only able to present the results of training LANPO with inter-sample feedback. **Results have shown that LANPO out-performs GRPO in three benchmarks especially the more challenging AIME-24/25, reinforcing LANPO's effectiveness in enhancing complex reasoning.** We shall update the remaining results as soon as the experiments finish.
>
> | Models | AIME25 | AIME24 | AMC23 | MATH500 | Avg |
> | --- | --- | --- | --- | --- | --- |
> | Qwen3-4B-Instruct-2507 | 43.4 | 54.47 | 77.73 | 95.31 | 67.72 |
> | SFT | 42.81 | 53.12 | 77.49 | 93.75 | 66.79 |
> | GRPO | 52.40 | 59.38 | 86.11 | 96.88 | 73.69 |
> | LANPO (w/ Inter-sample) | **55.42** | **60.42** | **87.58** | **96.88** | **75.08** |

---

> ### Author Response · Authors · 2025-11-23
> **Rebuttal Responses to Reviewer Wmpd (3/3)**
>
> > **Q4:** The “strong ICL benefit” in Figure 2 caption is unclear by reading the figure. Please consider using “inter-sample feedback”, or other appropriate terms.
> >
>
> **A4: Paper Update.** Thanks for the advice. In Figure 2, we adopt in-context examples (referred as ICL in the original manuscript) as language feedback at test-time to compare with its effect at training-time. We have revised the caption following your suggestion.
>
> ---
>
> > **Q5:** The first question is that whether the reflection and the contexts will be used to compute the gradient and back-propagated?
> >
>
> **A5:** To be short, **our forward/backward-pass treats the problems and the contexts as the states $s=(x, c)$, and** **the entire model responses as actions $a = y$** (including the reflections). Then we follow the standard GRPO procedure to compute advantage and policy loss for the state-action pairs $\pi(a | s)$. As a result, the context serves as conditions during gradient calculation, but we do not accumulate the token-loss from them. Such training familiars the model with the enhanced context, enabling it to reason with language feedback at test-time.
>
> ---
>
> > **Q6.1:** For the self-reflection process, what are the input and outputs?
> >
>
> **A6.1:** The input is the query and a previous response $(x, y)$, and the model is required to generate reflections on the correctness of $a$ and output a revised answer $y^\prime$ if needed.
>
> ---
>
> > **Q6.2:** What languages are added to the RL training context?
> >
>
> **A6.2:** There are two types of language feedback that we adopt during LANPO training, intra-sample and inter-sample feedback.
>
> - **For intra-sample feedback**, we provide historical responses to the same query, upon which the actor will reflect and generate new attempts.
> - **For inter-sample feedback**, we provide experiences summarized from successful solutions to similar problems, guiding the model to explore certain strategies.
>
> ---
>
> > **Q6.3:** Will there be a distribution shift if no intra-sample reflection is conducted during inference time?
> >
>
> **A6.3:** There is distribution shift between zero-shot inference and reasoning with language feedback, which is the exact reason why we mix the two reasoning modes during RL training, with the mix ratio controlled by the hyper-parameter $p$.
>
> ## **Thank You**
>
> Thank you once again for your review. We hope that our rebuttal addresses your concerns and we have updated the paper following your suggestions. We are eagerly looking forward to your kind responses.
>
> Best Regards,
>
> Authors

---

> > ### Comment · Reviewer_Wmpd · 2025-11-26
> > **Thanks for your response.**
> >
> > I first thank the authors for their informative responses, especially the experiments with small language model (QWen-4B). My concerns are mostly addressed. I decide to keep my positive score. This work proposes a novel way to incorporate natural language into RL training; and obtain decent improvement. I think this is a good contribution to the research community.
> >
> > But this work could be stronger if more efforts could be contributed to better explaining how these natural language feedbacks improve exploration, e.g., whether new reasoning capabilities have been learned [1].
> >
> > [1] Does Reinforcement Learning Really Incentivize Reasoning Capacity in LLMs Beyond the Base Model?

---

> > > ### Author Response · Authors · 2025-11-26
> > > **Follow-up Response to Reviewer Wmpd**
> > >
> > > Dear Reviewer Wmpd,
> > >
> > > Thank you for your positive assessment and the insightful comment about whether language feedback brings new reasoning abilities:
> > >
> > > > **Q: How these natural language feedbacks improve exploration, e.g., whether new reasoning capabilities have been learned?**
> > > >
> > >
> > > Following your question, **we would like to present two additional results supporting that LANPO teaches specific, learned skills.**
> > >
> > > ---
> > >
> > > **1. A Learned Skill of Self-Reflection**
> > >
> > > We first argue that effective self-reflection is precisely such a skill. To prove that LANPO specifically teaches this capability, we conducted a controlled experiment measuring the impact of a second-pass self-reflection on models trained with SFT, GRPO, and LANPO.
> > >
> > > | Model | AIME24 (Zero-shot) | AIME24 (Self-Reflect) | **Self-Reflection Gain** |
> > > | --- | --- | --- | --- |
> > > | Qwen2.5-7B-SFT | 10.83 | 9.90 | -0.93 |
> > > | Qwen2.5-7B-GRPO | 17.29 | 18.02 | +0.73 |
> > > | **Qwen2.5-7B-LANPO** | **20.42** | **22.19** | **+1.77** |
> > > | Qwen3-14B-SFT | 20.73 | 19.17 | -1.60 |
> > > | Qwen3-14B-GRPO | 47.40 | 50.94 | +3.54 |
> > > | **Qwen3-14B-LANPO** | 46.15 | **53.74** | **+7.59** |
> > >
> > > The results show LANPO is highly effective in teaching this capability.
> > >
> > > - SFT models regress with self-reflection, proving this skill is not innate or simply a result of longer context.
> > > - **LANPO models gain dramatically from self-reflection (+7.59 on Qwen3-14B), a direct result of our intra-sample feedback mechanism,** while GRPO makes marginal improvement under comparison.
> > >
> > > ---
> > >
> > > **2. A Learned Skill to Utilize External Guidance**
> > >
> > > LANPO also teaches the model to discern and use external feedback. Without our relevance filtering, providing retrieved experiences at test-time can be harmful. LANPO learns to turn this into a benefit.
> > >
> > > | Training Method | Zero-shot (Avg) | w/ Retrieval (Avg) | **Change from Zero-shot (Avg)** |
> > > | --- | --- | --- | --- |
> > > | w/o filtering | 44.64 | 44.11 | -0.53 |
> > > | **w/ filtering (ours)** | 44.14 | 44.83 | **+0.69** |
> > >
> > > **This result demonstrates a qualitative shift: the model learns to properly utilize external guidance,** a key skill for effective exploration for both training and testing.
> > >
> > > ---
> > >
> > > In summary, LANPO's improvements stem from instilling tangible reasoning skills such as self-reflection and leveraging experiences at test-time, which enables the model to explore beyond trail-and-error during training, thus bringing performance improvement.
> > >
> > > We sincerely appreciate your valuable guidance, and we hope the additional results can fully address your concerns.
> > >
> > > Best Regards,
> > >
> > > The Authors

---

> > > > ### Comment · Reviewer_Wmpd · 2025-11-26
> > > >
> > > > I thank the authors for providing these additional results. These results show that LANPO can further improve the GRPO performance. That said, in my opinion, they still do not sufficiently justify whether LANDPO introduces better explorations; or incentivizes new reasoning capabilities. I would recommend two ways to justify: 1, follow [1] and use pass@k performance against base mode and other baselines; 2, (a bit hand-wavy) qualitatively show new reasoning capabilities scaffolded by the self-reflection mechanism.
> > > >
> > > > It is also likely that I do not fully interpret these results. If so, I am happy to learn more of your insights.
> > > >
> > > > [1] Does Reinforcement Learning Really Incentivize Reasoning Capacity in LLMs Beyond the Base Model?

---

> ### Author Response · Authors · 2025-11-27
> **Futher Clarification on New Reasoning Capabilities**
>
> Dear Reviewer Wmpd,
>
> Thank you for the thoughtful follow-up and for directing us to the insightful discussion in [1]. We appreciate the opportunity to further clarify how LANPO incentivizes new reasoning capabilities, moving beyond simple performance gains.
>
> We agree with your distinction and the core point raised in [1]. We differentiate between:
>
> - **Reasoning Capacity [1]:** The theoretical upper bound of a model's reasoning potential, which, as [1] suggest, is difficult to expand significantly with standard RL fine-tuning.
> - **Reasoning Capability (our paper):** The demonstrated, practical ability to perform specific reasoning tasks, such as self-correction or utilizing external knowledge.
>
> Our work focuses on enhancing reasoning *capability*. The "base model > RL model" phenomenon noted in [1] is, as ProRL [2] shows, **most prominent in domains like math and coding that are already heavily saturated in pre-training data, but notably less significant in more OOD tasks**.
>
> [2] ProRL: Prolonged Reinforcement Learning Expands Reasoning Boundaries in Large Language Models
>
> ---
>
> **Instead of expanding the support space (i.e., reasoning capacity) of the RL policy, LANPO studies how to explore the existing support space more efficiently and throughly by teaching the model new, explicit reasoning *skills* through language feedback**. Our previous results demonstrate that LANPO does more than just improve zero-shot scores; **it instills tangible capabilities that are absent in the base model.**
>
> 1. **A Learned Skill of Self-Correction:** As shown in the table below, the base SFT model's performance *degrades* when asked to self-reflect. In contrast, the LANPO-trained model shows a dramatic improvement, gaining **+7.59 points** on Qwen3-14B. This is not a marginal gain; it is clear evidence that LANPO has taught the model a new, effective capability for self-correction that it did not previously possess.
> 2. **A Learned Skill to Utilize Experience:** Similarly, the model must learn how to effectively use external guidance. Without the relevance filtering taught by LANPO, providing retrieved experiences at test-time is actively harmful (a -0.53 drop). LANPO training transforms this into a benefit (+0.69 gain), demonstrating the model has acquired the capability to discern and apply relevant information.
>
> ---
>
> In response to your excellent suggestion, we would also like to provide a qualitative example of how LANPO's feedback mechanism scaffolds a new reasoning capability. In an example detailed in Appendix D, the model tackles a complex combinatorics problem about finding the number of 8-digit integers divisible by 22.
>
> During its case-by-case analysis, the model initially makes a counting error. However, prompted by the self-correction mechanism trained via LANPO, it performs an explicit verification step:
>
> > **Thought Flow Overview**
> >
> > 1. Problem Analysis
> >     - Identifies that divisibility by 22 requires meeting both divisibility by 2 and 11
> >     - Establishes two key conditions: even last digit and specific digit sum pattern
> > 2. Mathematical Framework
> >     - Calculates total digit sum
> >     - Derives that only possible valid case is when odd and even position sums both equal 18
> > 3. Case Analysis by Last Digit
> >     - Breaks problem into 4 cases based on possible even last digits:
> >     - For each case, finds 3-digit combinations from remaining digits
> > 4. Error Identification & Correction ⚡
> >     - Identifies discrepancy in Case 3 (last digit = 6): original solution claimed 5 combinations
> >     - Systematically re-evaluates combinations summing to 12
> >     - Corrects to 4 valid combinations
> >     - Adjusts calculation from 720 to 576 ways for this case
> > 5. Final Calculation
> >     - All four cases yield 576 valid arrangements each
> >     - Total valid integers: 4 × 576 = 2304
> >     - Final answer: 279
>
> Here, the model explicitly identifies the mistake, re-evaluates the logic, and documents the correction. This demonstrates a meta-cognitive reasoning skill—verifying the solution process itself—that goes beyond simple problem-solving and is a direct result of the capabilities instilled by LANPO's training process.
>
> ---
>
> Thank you again for pushing us to clarify this crucial point. If there is anything left for us to explain, please let us know!
>
> Best regards,
>
> Authors

---

### Official Review · Reviewer_7DfC · 2025-10-29

**Soundness:** 3
**Presentation:** 3
**Contribution:** 2
**Rating:** 2
**Confidence:** 2

**Summary:**

This paper proposes LANPO, a framework that integrates both language and numerical feedback into RL for LLMs. Traditional LLM-RL methods rely solely on scalar rewards, discarding valuable reasoning signals embedded in textual outputs. LANPO addresses this by introducing two key mechanisms: Reward-Agnostic Reflection, which enables safe intra-sample self-correction, and Relevant Abstraction, which filters and summarizes inter-sample experiences into transferable reasoning principles. Together, these mechanisms allow the model to reuse its own past rollouts as structured guidance during training. Experiments on multiple mathematical reasoning benchmarks (AIME24/25, AMC, MATH) demonstrate that LANPO consistently outperforms strong baselines such as GRPO across both 7B and 14B models, achieving stable optimization and better zero-shot generalization.

**Strengths:**

1. The paper addresses an underexplored yet important problem: how to effectively integrate language feedback into RL for LLMs, which is highly relevant to advancing reasoning and RL sample efficiency.
2. The paper introduces two well-motivated components: Reward-Agnostic Reflection, enabling safe and label-free intra-sample self-correction, and Relevant Abstraction, which distills inter-sample experiences into transferable principles for more robust generalization.
3. Consistent performance improvements: LANPO achieves notable gains over strong baselines such as GRPO across multiple reasoning benchmarks and model scales, under both 7B and 14B Qwen models.

**Weaknesses:**

1. My primary concern is that the experiments are conducted solely on Qwen models and mathematical reasoning tasks. Recent studies [1] have highlighted potential overfitting or evaluation bias in Qwen+math evaluations. The paper would be substantially strengthened by including results from additional models or domains.
2. LANPO introduces multiple auxiliary components (e.g., summarization, reflection, retrieval), which substantially increase computational cost and token consumption. The reported performance improvements may therefore be partially attributable to greater training time. A fairer comparison would normalize along the x-axis (training time or total token budget) and report performance on the y-axis.
3. LANPO updates the policy twice using the same data: once through parameter optimization and once through prompt modification (feedback injection). This effectively makes the training partially off-policy, since the prompt update alters the policy distribution between rollouts and updates. The authors do not discuss this issue, yet in practice, such prompt-level changes can substantially shift the policy distribution and degrade the stability of GRPO-like on-policy RL algorithms.
4. Recent works have shown that self-reflection mechanisms often fail to improve model outputs in the absence of external supervision signals [2]. A key conceptual concern remains: if the model can solve a problem correctly after reflection, why couldn’t it solve it correctly in the first pass? Is the improvement simply due to longer context or increased reasoning steps?
5. As shown in Table 2, the model attains 44.64 without filtering and 44.83 with filtering, indicating that filtering offers only a minor benefit when the testing configuration is consistent with the training setup.

[1] Spurious Rewards: Rethinking Training Signals in RLVR

[2] When Can LLMs Actually Correct Their Own Mistakes? A Critical Survey of Self-Correction of LLMs

**Questions:**

1. Have you evaluated LANPO on models or tasks beyond Qwen and math reasoning to test generalization?
2. Can you report results normalized by total training tokens or compute to ensure fairness?
3. Does updating prompts between rollouts introduce off-policy effects or instability?
4. Are the reflection gains due to better reasoning or simply longer context?

---

> ### Author Response · Authors · 2025-11-23
> **Rebuttal Responses to Reviewer 7dfc (1/5)**
>
> Dear Reviewer 7DfC:
>
> We would like first appreciate you for acknowledging the **motivation and performance** of our proposed method, LANPO.  Following your insightful comments on our experimental setting, the computational overhead, and the ablations, we have prepared additional results and discussions below, with which we hope can address your concerns. Should you have any further questions, please feel free to raise them; all your feedback will help us improve our work.
>
> ---
>
> > **Q1.** My primary concern is that the experiments are conducted solely on Qwen models and mathematical reasoning tasks. Recent studies [1] have highlighted potential overfitting or evaluation bias in Qwen+math evaluations. The paper would be substantially strengthened by including results from additional models or domains.
> >
>
> > Have you evaluated LANPO on models or tasks beyond Qwen and math reasoning to test generalization?
> >
>
> **A1:** We acknowledge you for raising this critical point. With [1] mainly raises concern about the Qwen2.5 + MATH setting, we have broaden the scope of our experiment to the latest model (Qwen3-4B-Instruct-2507), and the coding task, with results listed below.
>
> **New Results on the Latest Model.** To broaden the scope of our study, we conduct additional experiments with Qwen3-4B-Instruct-2507 on the math reasoning tasks. **Results have shown that LANPO out-performs GRPO in three benchmarks especially the more challenging AIME-24/25, reinforcing LANPO's effectiveness in enhancing complex reasoning.**
>
> | Models | AIME25 | AIME24 | AMC23 | MATH500 | Avg |
> | --- | --- | --- | --- | --- | --- |
> | Qwen3-4B-Instruct-2507 | 43.4 | 54.47 | 77.73 | 95.31 | 67.72 |
> | SFT | 42.81 | 53.12 | 77.49 | 93.75 | 66.79 |
> | GRPO | 52.40 | 59.38 | 86.11 | 96.88 | 73.69 |
> | LANPO |  **55.42** | **60.42** | **87.58** | **96.88** | **75.08** |
>
> **Results on Coding Task with Qwen2.5-7B.** We agree with you that the generalizability of LANPO should be broadly tested. Following your question, We have conducted an extra set of experiments with Qwen2.5-7B on the coding dataset Eurus2 [1]. The results are provided in the table below. The most effective variant, LANPO with both feedback types, achieves the highest average score of 35.91, **an absolute improvement of 2.75 points over the GRPO baseline. All LANPO variants outperform GRPO on average and on the TACO and CodeContests benchmarks.** We note a minor exception where LANPO with only intra-sample feedback slightly underperforms GRPO on the CodeForces test set, but the overall trend confirms LANPO's robust performance in the coding domain.  We are currently working on incorporating more application scenarios of LANPO and will update the results as soon as we could.
>
> | Coding | CodeForces | CodeTACO | CodeContests | Avg |
> | --- | --- | --- | --- | --- |
> | Qwen2.5-7B-Instruct | 8.54 | 8.15 | 18.87 | 11.85 |
> | SFT | 9.25 | 8.02 | 15.32 | 10.86 |
> | GRPO | 40.04 | 22.28 | 37.17 | 33.16 |
> | LANPO w/ Inter-Sample Feedback | **42.38** | 23.13 | 38.46 | 34.16 |
> | LANPO w/ Intra-Sample Feedback | 39.74 | 23.23 | 38.56 | 33.84 |
> | LANPO w/ Both Feedback | 41.77 | **25.23** | **40.72** | **35.91** |
>
> **Paper Update:** Following your advice, we have incorporated the results as well as discussions on Qwen3-4B into our revised manuscript as Table 5 in Appendix A.4.  And the results on the coding task are can be found in Table 6 ( Appendix A.5).
>
> [1] https://huggingface.co/datasets/PRIME-RL/Eurus-2-RL-Data

---

> ### Author Response · Authors · 2025-11-23
> **Rebuttal Responses to Reviewer 7dfc (2/5)**
>
> > **Q2:** LANPO introduces multiple auxiliary components (e.g., summarization, reflection, retrieval), which **substantially increase computational cost** and **token consumption.** The reported performance improvements may therefore be partially attributable to **greater training time.** A fairer comparison would normalize along the x-axis (training time or total token budget) and report performance on the y-axis.
> >
>
> > Can you report results normalized by total training tokens or compute to ensure fairness?
> >
>
> **A2: Breakdown of Training Step Time:** The primary overhead of LANPO during training comes from processing longer input sequences when language feedback is included. We profiled a training step for GRPO vs. LANPO on a Qwen2.5-7B model, and the results are below:
>
> | Training Stage | GRPO | LANPO (inter-sample) | LANPO (intra-sample) |
> | --- | --- | --- | --- |
> | **Rollout** | 222s (30.62%) | 328s (23.05%) | 346s (23.96%) |
> | **Calculating Old/Ref Log-Prob** | 150s (20.69%) | 324s (22.77%) | 314s (21.75%) |
> | **Reward Calculation** | 8s (1.10%) | 8s (0.56%) | 8s (0.55%) |
> | **Weight Update** | 342s (47.17%) | 699s (49.12%) | 708s (49.03%) |
> | **Other Overhead (communication, experience pool update etc.)** | 3s (0.41%) | 64s (4.50%) | 68s (4.71%) |
> | **End to End** | **725s** | **1423s** | **1444s** |
> | **Total Training Time** | **~72 hours** | **~146 hours** | **~138 hours** |
>
> **Performance vs. Compute Budget:** A critical question is whether LANPO's gains are simply due to this increased computation. To investigate this, we trained the GRPO baseline for more steps (730 vs. 330) to closely match the total training time of LANPO (~160 hours vs. ~146 hours).
>
> | Qwen2.5 7B | AIME25 | AIME24 | Total Training Time | Total Input Tokens | Total Output Tokens |
> | --- | --- | --- | --- | --- | --- |
> | GRPO-330 steps | 13.02 | 17.29 | ~72 hours | ~0.04M | ~0.5M |
> | GRPO-730 steps | 15.9 | 18.6 | ~160 hours | ~0.08M | ~1M |
> | LANPO-330 steps (w/ Inter-sample) | **16.04** | **19.48** | ~146 hours | ~0.2M | ~0.6M |
>
> The results demonstrate that **LANPO outperforms the compute-matched GRPO baseline, especially on the more difficult AIME24 benchmark.** This, along with the training curves in Figure 5(a) showing LANPO's faster reward increase in terms of training steps, confirms that LANPO's performance gains stem from improved sample efficiency and more effective learning, not just from consuming more FLOPs.
>
> **Future Work on Acceleration:** In this work, we mainly aim to study whether incorporating language feedback into numerical RL has the potential to outperform the classic trial-and-error loop. Thus, we focus more on the comparison in terms of sample complexity, i.e., how many state-action-reward triplets are used to learn the policy, rather than the empirical time cost. We agree wall-clock efficiency is a critical aspect for the practical application of RL methods. For exciting avenues of optimization, one can experiment with condensing the experiences into shorter formats, or adopting shorter system prompts, which could further reduce the computational overhead and make LANPO even more practical.
>
> **Paper Update:** Following your suggestions, we have incorporated concrete discussions and results about the computational efficiency of LANPO into the manuscript as Table 7 and Table 8 in Appendix A.

---

> ### Author Response · Authors · 2025-11-23
> **Rebuttal Responses to Reviewer 7dfc (3/5)**
>
> > **Q3:** LANPO updates the policy twice using the same data: once through parameter optimization and once through prompt modification (feedback injection). This effectively makes the training partially off-policy, since the prompt update alters the **policy distribution** between rollouts and updates. The authors **do not discuss this issue**, yet in practice, such prompt-level changes can substantially shift the policy distribution and degrade the stability of GRPO-like on-policy RL algorithms.
> >
>
> > Does updating prompts between rollouts introduce **off-policy effects or instability?**
> >
>
> **A3: Stability and On-Policy of LANPO.** We thank the reviewer for this perceptive question about the on-policy nature of LANPO, but we argue that LANPO remains stable and methodologically sound for the following reasons:
>
> 1. **LANPO is On-Policy Curriculum RL:** First, our introduction of language feedback does not alter on-policy sampling, with **the policy gradient always being calculated on the trajectories sampled from the latest iteration**. In fact, updating prompt distribution together with policy optimization is a form of curriculum RL [1], which has received wide attention for RL in LLM reasoning such as [2] and [3]. Rather than having a fixed input distribution, LANPO adapts the prompt based on the agent's recent experience. This is a deliberate design choice to improve exploration and is a well-established paradigm for enhancing RL.
> 2. **Explicit Control via Hybrid Training:** Second, we anticipated that a pure feedback-driven RL could create a distribution shift between training and zero-shot inference. This is precisely why we introduced the feedback ratio hyper-parameter $p_t$. Ablations into the hyper-parameter (Section 5.2) shows that appropriate balance between the two exploration modes boosts performance ( $p_t=0.5$).
> 3. **Empirical Evidence of Stability:** Crucially, our training dynamics provide direct evidence against instability. Figure 5(b) in our paper shows that the gradient norms for all LANPO variants are smooth and stable, comparable to the GRPO baseline. This empirically demonstrates that our method does not degrade the stability of the underlying on-policy algorithm.
>
> In summary, LANPO's language loop is a principled form of curriculum learning that enhances exploration, while its numerical loop and hybrid training design maintain the stability of the on-policy RL framework.
>
> [1] Curriculum Learning for Reinforcement Learning Domains: A Framework and Survey; https://arxiv.org/abs/2003.04960
>
> [2] Self-Evolving Curriculum for LLM Reasoning; https://arxiv.org/abs/2505.14970
>
> [3] Curriculum Reinforcement Learning from Easy to Hard Tasks Improves LLM Reasoning; https://arxiv.org/abs/2506.06632

---

> ### Author Response · Authors · 2025-11-23
> **Rebuttal Responses to Reviewer 7dfc (4/5)**
>
> > **Q4.1:** Recent works have shown that self-reflection mechanisms often fail to improve model outputs in the absence of external supervision signals [2]. A key conceptual concern remains: if the model can solve a problem correctly after reflection, why couldn’t it solve it correctly in the first pass? Is the improvement simply due to longer context or increased reasoning steps? Are the reflection gains due to better reasoning or simply longer context?
> >
>
> We thank the reviewer for raising this fundamental question about the mechanism of self-correction. Our response is two-fold: first, we provide a conceptual model for why a second pass can succeed where the first fails, and second, we provide clear empirical evidence that this ability is learned through LANPO, not an artifact of longer context.
>
> **A4.1: Conceptual Model, Reasoning as Search and the Solver-Verifier Gap.** The question of can be understood by viewing complex reasoning as **a search through a vast tree of possible reasoning steps**. An auto-regressive LLM generates a single path through this tree, often in a greedy fashion. A single mistake early on can lead it down an unrecoverable path.
> **A second pass (self-reflection) allows the model to "backtrack" and re-evaluate its initial path.** This is effective due to the "solver-verifier gap" [1]: a model is often better at recognizing a mistake in a completed solution (verifying) than it is at avoiding that mistake during initial generation (solving). Self-reflection provides the opportunity to leverage this stronger verification ability to find and correct flaws.
>
> **A4.2: The Ability to Self-Correct is a Learned Skill.** To prove that LANPO teaches this skill and that the gains are not merely from longer context, we conducted a controlled experiment. We compare the effect of a second-pass self-reflection on models trained with SFT, GRPO, and our LANPO. If the "longer context" hypothesis were true, all models should benefit similarly. Our results show the opposite:
>
> | Models | AIME24 (zero-shot) | AIME24 (Self-Reflect) | Self-Reflection Gain |
> | --- | --- | --- | --- |
> | Qwen2.5-7B-SFT | 10.83 | 9.9 | -0.93 |
> | Qwen2.5-7B-GRPO | 17.29 | 18.02 | +0.73 |
> | Qwen2.5-7B-LANPO (w/ Intra-sample) | **20.42** | **22.19** | **+1.77** |
> | Qwen3-14B-SFT | 20.73 | 19.17 | -1.6 |
> | Qwen3-14B-GRPO | **47.40** | 50.94 | +3.54 |
> | Qwen3-14B-LANPO (w/ Intra-sample) | 46.15 | **53.74** | **+7.59** |
>
> This data leads to a clear conclusion:
>
> - Naive SFT models cannot self-correct effectively; giving them a second chance actually hurts performance. This refutes the idea that gains come from longer context alone.
> - **LANPO-trained models show dramatically larger gains from self-correction compared to both SFT and standard GRPO baselines.** This is because our reward-agnostic intra-sample feedback mechanism directly trains the model to critique and refine its own reasoning paths during the RL process.
>
> In summary, the improvement is not simply due to increased reasoning steps. It is due to a learned skill of effective self-correction, which LANPO's intra-sample feedback mechanism is specifically designed to teach.
>
> [1] A Theoretical Understanding of Self-Correction through In-context Alignment; https://arxiv.org/abs/2405.18634; NeurIPS 2024
>
> [2] When Can LLMs Actually Correct Their Own Mistakes? A Critical Survey of Self-Correction of LLMs; https://aclanthology.org/2024.tacl-1.78/; ACL 2024
>
> [3] Training Language Models to Self-Correct via Reinforcement Learning; https://proceedings.iclr.cc/paper_files/paper/2025/file/871ac99fdc5282d0301934d23945ebaa-Paper-Conference.pdf; ICLR 2025
>
> **Paper Update:** Thanks for the in-depth question, we have revised Appendix A.2 to further discuss the comparison between GRPO and LANPO in terms of using intra-sample feedback at test-time.
>
> ---

---

> ### Author Response · Authors · 2025-11-23
> **Rebuttal Responses to Reviewer 7dfc (5/5)**
>
> > **Q5:** As shown in Table 2, the model attains 44.64 without filtering and 44.83 with filtering, indicating that filtering offers only a minor benefit when the testing configuration is consistent with the training setup.
> >
>
> We thank you for raising this point, as it allows us to clarify the core function of the relevance filter.
>
> **A5: Relevance Filtering Enables Reasoning With Experiences at Test-Time.** The filter's primary role is not to improve zero-shot performance, but to enable the model to effectively reason with retrieved experiences at test-time. This is demonstrated by comparing the 'Zero-shot' vs. 'w/ Retrieval' modes for different training methods in Table 3 of our paper (quoted below):
>
> | Training Method | Zero-shot (AIME25) | w/ Retrieval (AIME25) | Change from Zero-shot (AIME25) | Zero-shot (Avg) | w/ Retrieval (Avg) | Change from Zero-shot (Avg) |
> | --- | --- | --- | --- | --- | --- | --- |
> | w/o filtering | 15.21 | 13.65 | -1.56 | 44.64 | 44.11 | -0.53 |
> | w/ filtering (ours) | 16.04 | 16.98 | +0.94 | 44.14 | 44.83 | +0.69 |
>
> **Without relevance filtering, providing experiences at test time hurts performance**. Crucially, this result demonstrates a qualitative shift in the model's capability. Without filtering, providing experiences at test-time is actively harmful, causing a -0.53 drop in performance. With filtering, the model learns to properly utilize these experiences, turning a detriment into a benefit (+0.69 gain). The key contribution is not the specific magnitude of this gain, but the fact that our method successfully unlocks this mode of reasoning, which is difficult with naive training
>
> **Paper Update:** We recognize that this point was not clear enough in the original text. We have revised Section 5.2 to more explicitly state that the primary effectiveness of the relevance filter is to enable effective test-time augmentation.
>
> ---
>
> ## **Thank You**
>
> Thank you once again for your review. We hope that our rebuttal addresses your concerns and we have updated the paper following your suggestions,. We are eagerly looking forward to your kind responses.
>
> Best Regards,
>
> Authors

---

> > ### Comment · Reviewer_7DfC · 2025-11-25
> >
> > Thanks for the additional experiments and clarifications. They addressed my concerns 1, 2, 4, and 5, and I have raised my score to 4.
> >
> > Regarding Q3, I still disagree with the authors’ interpretation. The “context feedback” mechanism cannot be explained as curriculum learning: curriculum typically adjusts inputs *smoothly*, whereas here the context can change the prompt *drastically*.
> >
> > Your observation about the necessity of $p_t$ for stable training further confirms that on-policy considerations are crucial. When $p_t = 1$, the in-context feedback shifts the data distribution too aggressively, causing a mismatch with the sampling policy and destabilizing the on-policy GRPO update.
> >
> > Overall, the paper provides a workable solution for combining numerical and language feedback. It is a solid paper, though the core challenge remains unaddressed: numerical-learning updates are slow but low-bias/high-variance, while in-context learning is fast but biased. The deeper question is when the model should rely on in-context learning/numerical feedback, and how to combine the two learning processes operating at different temporal scales so that we exploit their complementary strengths. In its current form, $p_t$ seems to inherit limitations from both sides, which is why I am not giving a higher score.

---

> > > ### Author Response · Authors · 2025-11-26
> > > **Follow-up Response to Reviewer 7DfC**
> > >
> > > Dear Reviewer 7DfC,
> > >
> > > Thank you very much for your detailed follow-up and for raising your score. We are grateful for **your positive assessment that our paper provides a "workable solution" and is "solid."**
> > >
> > > We also appreciate the deeper, more fundamental questions you've raised regarding the interaction between numerical and language-based learning. We would like to offer further clarification on how LANPO is designed to address it, particularly in response to your remaining concern about on-policy stability (Q3).
> > >
> > > ---
> > >
> > > > **Q6:** Regarding Q3, I still disagree with the authors’ interpretation. The “context feedback” mechanism cannot be explained as curriculum learning: curriculum typically adjusts inputs smoothly, whereas here the context can change the prompt drastically. Your observation about the necessity of $p_t$ for stable training further confirms that on-policy considerations are crucial.
> > > >
> > >
> > > **A6:** We respectfully argue that the language feedback mechanism in LANPO is designed to be inherently smooth and coupled with the policy's evolution, preventing the kind of drastic, destabilizing shifts you mentioned. This smoothness arises from two key design choices:
> > >
> > > 1. **The Experience Pool is On-Line and Evolving:** The experience pool, from which language feedback is generated, is not a fixed, external dataset. It is accumulated **on-line** from the policy's own recent trajectories. As the policy gradually improves through GRPO updates, the quality and nature of the experiences (both successes and failures) entering the pool also evolve smoothly. This ensures that the feedback (e.g., summaries of successful strategies) always reflects the policy's current capabilities, preventing drastic shifts caused by out-of-distribution information. The curriculum is thus self-generated and paced by the agent's own progress.
> > > 2. **Shared Parameters Minimize Divergence:** All modules—the summarizer, reflector, and the policy model itself—**share the same underlying set of parameters**. This is a crucial design choice to ensure that the "mind" generating the feedback is the same "mind" that is being trained. As the policy parameters are updated via the numerical GRPO step, the capabilities of the feedback-generation modules also co-evolve. **This tight coupling minimizes the divergence between the policy's behavior and the feedback it receives**, further contributing to a stable and progressive learning curriculum rather than an abrupt, external intervention.
> > >
> > > ---
> > >
> > > > **Q7:** The deeper question is when the model should rely on in-context learning/numerical feedback, and how to combine the two learning processes operating at different temporal scales so that we exploit their complementary strengths. In its current form, LANPO seems to inherit limitations from both sides.
> > > >
> > >
> > > **A7:** We agree on your summary of the core challenge, but we would like to emphasize that LANPO's design is not a simple mixture but a **synergistic, bootstrapping process** where the two loops are designed to be mutually beneficial.
> > >
> > > - The slow, low-bias **numerical updates (GRPO)** ground the model in robust, reward-driven learning, gradually improving its core reasoning ability. This improved ability, in turn, allows the model to generate **higher-quality language feedback** (better reflections, more insightful summaries of its own traces).
> > > - Conversely, **language feedback (the 'fast' loop)** provides richer, more informative learning signals than a single scalar reward. It helps the model overcome the sample inefficiency of RL by explicitly pointing out flaws or successful patterns, thus **accelerating the 'slow' numerical learning process**. As we demonstrate in our compute-matched experiments (Table 8), LANPO achieves better results than a longer-trained GRPO, which supports this claim of accelerated, more efficient learning.
> > >
> > > **In essence, language feedback refines the exploration space for numerical RL, while numerical RL validates and strengthens the reasoning patterns suggested by language feedback**. The $p_t$ hyper-parameter, which we agree to be vital, is the explicit lever to manage this balance. We believe this bootstrapping dynamic, further detailed in Section 4 of our revised paper, is a principled approach to combining these two learning paradigms, where each loop mitigates the other's weaknesses rather than inheriting them.
> > >
> > > ---
> > >
> > > Thank you once again for your constructive feedback. Your comments have helped us sharpen our understanding into the interplay between language and numerical feedback in RL. With further explanation clarifing our design philosophy and the synergistic nature of LANPO, we hope all your concerns can be addressed.
> > >
> > > Best Regards,
> > >
> > > Authors

---

### Official Review · Reviewer_yu24 · 2025-11-02

**Soundness:** 2
**Presentation:** 3
**Contribution:** 1
**Rating:** 4
**Confidence:** 4

**Summary:**

This paper proposed LANPO, a framework that incorporates language feedback in RL to guide exploration. LANPO builds and experience pool to provide 2 types of feedback:

1. Reward agnostic reflection where the model is is shown its prior response to a prompt and asked to critique it before producing a refined response. This is different from naive intra-sample feedback where the model is shown the label instead. By making intra-sample feedback reward agnostic, LANPO prevents the label leakage issue.

2. Relevant abstraction for inter sample feedback. To prevent the behavior collapse induced by naively incorporating raw solutions, LANPO first finds sufficiently similar trajectories from the experience pool and then condenses them into high-level principles that generalize across problems.

During RL, the model alternates between from scratch and feedback aware rollouts to ensure that the model can perform well with and without drawing from the experience pool during inference.

LANPO consistently outperforms GRPO on sample efficiency and achieves an absolute performance improvement of 9.27% on the AIME25 test set after the same number of training steps.

**Strengths:**

- The paper introduces a novel and thoughtful way to incorporate language feedback into RL. Utilizing the experience pool to provide two different types of feedback allows for the model to learn introspection and how to generalize transferable learnings across examples.
 -  Intra-example feedback in particular consistently improves zero-shot and self-corrected performance across all benchmarks even without language feedback training.
- LANPO achieves better sample efficiency than GRPO, while improving both zero-shot and feedback augmented inference.
 - The paper is well written with a number of insightful ablations that study the impact of filtering on inter-example feedback, and the balance between feedback aware rollouts and from scratch generation during RL,

**Weaknesses:**

- LANPO's effectiveness is studied in a relatively limited setting. Only Qwen 7B and 14B models were included in all experiments and all the experiments were conducted on math tasks. For better generalizability, it would be helpful to see results on a broader range of model sizes. It's interesting that the 7B model benefits more from LANPO than the 14B model, being able to study any trends here would be nice.

- Including experiments with LANPO on at least one other type of tasks like coding, safety, general instruction following etc would go further in proving its generalizability. It could be difficult to find relevant examples from the experience pool for broader tasks complicating the relevant abstraction step.

- The computational overhead introduced by LANPO should be discussed further. In particular, the relevance calculation requires an LLM inference which would add significant overhead during training. The paper would be improved with precise numbers on step time increases introduced by each part of the pipeline.

- The relevance filtering ablation studies the effect of no filtering vs. $\gamma = 0.9$ threshold with mixed results. AIME 24 and MATH and better without filtering whereas AIME25 and AMC are better with filtering. This indicates inconsistent utility of filtering.

**Questions:**

- How many prompts with feedback are truncated during training? Similarly how many intra example feedback rollouts are truncated?

---

> ### Author Response · Authors · 2025-11-23
> **Rebuttal Responses to Reviewer yu24 (1/3)**
>
> Dear Reviewer yu24:
>
> We would like first appreciate you for acknowledging the **novelty and performance** of our proposed method, LANPO.  Following your insightful comments on our experimental setting, the computational overhead, and the ablations, we have prepared additional results and discussions below, with which we hope can address your concerns.  Should you have any further questions, please feel free to raise them; all your feedback will help us improve our work.
>
> ---
>
> > **Q1:** LANPO's effectiveness is studied in a relatively limited setting. Only Qwen 7B and 14B models were included in all experiments and all the experiments were conducted on math tasks. For better generalizability, it would be helpful to see results on a broader range of model sizes. It's interesting that the 7B model benefits more from LANPO than the 14B model, being able to study any trends here would be nice.
> >
>
> **A1: New Results on a Smaller Model (Qwen3-4B).** To broaden the scope of our study, we conduct additional experiments with Qwen3-4B-Instruct-2507 on the math reasoning tasks. Due to the tight time constraint of the rebuttal period, we are currently only able to present the results of training LANPO with inter-sample feedback. **Results have shown that LANPO out-performs GRPO in three benchmarks especially the more challenging AIME-24/25, reinforcing LANPO's effectiveness in enhancing complex reasoning.** We shall update the remaining results as soon as the experiments finish.
>
> | Models | AIME25 | AIME24 | AMC23 | MATH500 | Avg |
> | --- | --- | --- | --- | --- | --- |
> | Qwen3-4B-Instruct-2507 | 43.4 | 54.47 | 77.73 | 95.31 | 67.72 |
> | SFT | 42.81 | 53.12 | 77.49 | 93.75 | 66.79 |
> | GRPO | 52.40 | 59.38 | 86.11 | 96.88 | 73.69 |
> | LANPO | **55.42** | **60.42** | **87.58** | **96.88** | **75.08** |
>
> **Clarification on Performance Gains for 7B vs. 14B Models.**  We appreciate the reviewer's observation about the relative gains between the 7B and 14B models. **We would like to clarify that LANPO provides substantial benefits to the 14B model as well**, supported by the table below (quoted from Table 1). The performance gain is especially significant on AIME25, increasing from 33.02 to 38.23 under zero-shot testing, which is more significant than Qwen2.5-7B. In practice, our hyper-parameter tuning were conducted with 7B models and directly transferred to 14B models. This practical approach may result in suboptimal performance for the 14B model, **yet LANPO still demonstrates strong improvements**. We believe that with dedicated tuning for the 14B model, the gains could be even more pronounced.
>
> | Qwen3-14B | GRPO | LANPO w/ Intra-Sample Feedback | LANPO w/ Intra-Sample Feedback + Test-time Feedback |
> | --- | --- | --- | --- |
> | AIME-25 | 33.02 | 38.23 | **42.29** |
> | Avg. Acc. | 62.66 | 64.02 | **67.83** |
>
> **Paper Update:** We have incorporated the results as well as discussions on Qwen3-4B into our revised manuscript as Table 5 in Appendix A.4.
>
> ---
>
> > **Q2:** Including experiments with LANPO on at least one other type of tasks like coding, safety, general instruction following etc would go further in proving its generalizability. It could be difficult to find relevant examples from the experience pool for broader tasks complicating the relevant abstraction step.
> >
>
> **A2: Results on Coding Task with Qwen2.5-7B.** We agree with you that the generalizability of LANPO should be broadly tested. Following your question, We have conducted an extra set of experiments with Qwen2.5-7B on the coding dataset Eurus2 [1]. The results are provided in the table below. The most effective variant, LANPO with both feedback types, achieves the highest average score of 35.91, **an absolute improvement of 2.75 points over the GRPO baseline. All LANPO variants outperform GRPO on average and on the TACO and CodeContests benchmarks.** We note a minor exception where LANPO with only intra-sample feedback slightly underperforms GRPO on the CodeForces test set, but the overall trend confirms LANPO's robust performance in the coding domain.  We are currently working on incorporating more application scenarios of LANPO and will update the results as soon as we could.
>
> | Coding | CodeForces | CodeTACO | CodeContests | Avg |
> | --- | --- | --- | --- | --- |
> | Qwen2.5-7B-Instruct | 8.54 | 8.15 | 18.87 | 11.85 |
> | SFT | 9.25 | 8.02 | 15.32 | 10.86 |
> | GRPO | 40.04 | 22.28 | 37.17 | 33.16 |
> | LANPO w/ Inter-Sample Feedback | **42.38** | 23.13 | 38.46 | 34.16 |
> | LANPO w/ Intra-Sample Feedback | 39.74 | 23.23 | 38.56 | 33.84 |
> | LANPO w/ Both Feedback | 41.77 | **25.23** | **40.72** | **35.91** |
>
> **Paper Update:** Following your advice, we have incorporated the results on the coding task into our revised manuscript as Table 6 in Appendix A.5.
>
> [1] https://huggingface.co/datasets/PRIME-RL/Eurus-2-RL-Data

---

> ### Author Response · Authors · 2025-11-23
> **Rebuttal Responses to Reviewer yu24 (2/3)**
>
> > **Q3:** The computational overhead introduced by LANPO should be discussed further. In particular. The paper would be improved with precise numbers on step time increases introduced by each part of the pipeline. The relevance calculation requires an LLM inference which would add significant overhead during training.
> >
>
> **A3: Clarification on Relevance Calculation.** First, we would like to clarify a key implementation detail that mitigates a major potential bottleneck. We are aware of that enumerating every problem-experience pair online is highly inefficient. As a result, **we pre-compute and store the relevance scores between problems before training, enabling fast online retrieval**, which we have mentioned in Line 355-356 in the submitted manuscript.
>
> **Breakdown of Training Step Time:** The primary overhead of LANPO during training comes from processing longer input sequences when language feedback is included. We profiled a training step for GRPO vs. LANPO on a Qwen2.5-7B model, and the results are below:
>
> | Training Stage | GRPO | LANPO (inter-sample) | LANPO (intra-sample) |
> | --- | --- | --- | --- |
> | **Rollout** | 222s (30.62%) | 328s (23.05%) | 346s (23.96%) |
> | **Calculating Old/Ref Log-Prob** | 150s (20.69%) | 324s (22.77%) | 314s (21.75%) |
> | **Reward Calculation** | 8s (1.10%) | 8s (0.56%) | 8s (0.55%) |
> | **Weight Update** | 342s (47.17%) | 699s (49.12%) | 708s (49.03%) |
> | **Other Overhead (communication, experience pool update etc.)** | 3s (0.41%) | 64s (4.50%) | 68s (4.71%) |
>
> **Performance vs. Compute Budget:** A critical question is whether LANPO's gains are simply due to this increased computation. To investigate this, we trained the GRPO baseline for more steps (730 vs. 330) to closely match the total training time of LANPO (~160 hours vs. ~146 hours).
>
> | Qwen2.5 7B | AIME25 | AIME24 | Total Training Time | Total Input Tokens | Total Output Tokens |
> | --- | --- | --- | --- | --- | --- |
> | GRPO-330 steps | 13.02 | 17.29 | ~72 hours | ~0.04M | ~0.5M |
> | GRPO-730 steps | 15.9 | 18.6 | ~160 hours | ~0.08M | ~1M |
> | LANPO-330 steps (w/ Inter-sample) | **16.04** | **19.48** | ~146 hours | ~0.2M | ~0.6M |
>
> The results demonstrate that **LANPO outperforms the compute-matched GRPO baseline, especially on the more difficult AIME24 benchmark.** This, along with the training curves in Figure 5(a) showing LANPO's faster convergence in terms of training steps, confirms that LANPO's performance gains stem from improved sample efficiency and more effective learning, not just from consuming more FLOPs.
>
> **Future Work on Acceleration:** In this work, we mainly aim to study whether incorporating language feedback into numerical RL has the potential to outperform the classic trial-and-error loop. Thus, we focus more on the comparison in terms of sample complexity, i.e., how many state-action-reward triplets are used to learn the policy, rather than the empirical time cost. We agree wall-clock efficiency is a critical aspect for the practical application of RL methods. For exciting avenuess of optimization, one can experiment with condensing the experiences into shorter formats, or adopting shorter system prompts, which could further reduce the computational overhead and make LANPO even more practical.
>
> **Paper Update:** Following your suggestions, we have incorporated concrete discussions and results about the computational efficiency of LANPO into the manuscript as Table 7 and Table 8 in Appendix A.

---

> ### Author Response · Authors · 2025-11-23
> **Rebuttal Responses to Reviewer yu24 (3/3)**
>
> > **Q4:** The relevance filtering ablation studies the effect of no filtering vs. threshold with mixed results. AIME 24 and MATH and better without filtering whereas AIME25 and AMC are better with filtering. This indicates inconsistent utility of filtering.
> >
>
> **A4: Relevance Filtering Enables Reasoning With Experiences at Test-Time.** The core function of the relevance filter during training is to teach the model how to properly use inter-sample feedback. An unfiltered stream of experiences contains irrelevant noise, which prevents the model from learning this skill. **The success of this approach is not best measured by zero-shot performance, but by the model's performance when provided with retrieved experiences at inference.**
>
> | Training Method | Zero-shot (AIME25) | w/ Retrieval (AIME25) | Change from Zero-shot (AIME25) | Zero-shot (Avg) | w/ Retrieval (Avg) | Change from Zero-shot (Avg) |
> | --- | --- | --- | --- | --- | --- | --- |
> | w/o filtering | 15.21 | 13.65 | -1.56 | 44.64 | 44.11 | -0.53 |
> | w/ filtering (ours) | 16.04 | 16.98 | +0.94 | 44.14 | 44.83 | +0.69 |
>
> Without relevance filtering, providing experiences at test time hurts performance. On the contrary, training with filtering enables the model to successfully leverage relevant experiences at test-time, bringing a +0.94 performance gain on AIME-25 and +0.69 on average.
>
> **Variation in Zero-shot Performance:** The variations in zero-shot performance are a secondary effect. Training without a filtering mechanism mimics fast convergence like LANPO at early training stage, but eventually collapses into naive trial-and-error since all experiences are ignored by the policy (examples in Appendix A.1). In summary, the filtering mechanism is not inconsistent. It is our effort to prevent "behavior collapse" and imbue the model with the ability to use retrieved experiences, a core goal of our LANPO framework.
>
> **Paper Update:** We recognize that this point was not clear enough in the original text. We have revised Section 5.2 to more explicitly state that the primary effectiveness of the relevance filter is to enable effective test-time augmentation.
>
> ---
>
> > **Q5:** How many prompts with feedback are truncated during training? Similarly how many intra example feedback rollouts are truncated?
> >
>
> **A5: Ratio of Truncated Prompts and Rollouts.** The ratio of the truncated prompts with feedback is well below 5% in our training runs as we adopt a 3K maximum prompt length (Table 7 in the Appendix). The ratio of the truncated rollouts varies between 0% to 10% in most non-collapsed training runs, often reaching around 3% in the final stage of training. In general, extracting and using language feedback obtained from rollouts does not incur large portion of prompts/rollouts being truncated.
>
> ---
>
> ## **Thank You**
>
> Thank you once again for your review. We hope that our rebuttal addresses your concerns and we have updated the paper following your suggestions,. We are eagerly looking forward to your kind responses.
>
> Best Regards,
>
> Authors

---

> > ### Author Response · Authors · 2025-11-27
> > **Follow-up Response to Review yu24**
> >
> > Dear Reviewer yu24,
> >
> > We hope this message finds you well. We first deeply appreciate the time and effort you invested in your thorough review. Your feedback has been invaluable in helping us strengthen our work.
> >
> > Following your comments, we have posted our detailed rebuttal and have since updated our manuscript. We wanted to briefly summarize our responses to your insightful questions:
> >
> > Our paper introduces LANPO, a framework that leverages language feedback to guide exploration in reinforcement learning through two novel mechanisms: reward-agnostic reflection for intra-sample feedback and relevant abstraction for inter-sample feedback.
> >
> > In our rebuttal, we have sought to address your primary concerns:
> >
> > 1. **Generalizability:** To address the limited experimental setting, we have conducted new experiments on a smaller **Qwen3-4B model** for the math tasks and on a **new coding task (Eurus2)**. Both sets of results demonstrate that LANPO consistently outperforms the GRPO baseline, reinforcing its effectiveness and generalizability. We also clarified that the performance gains on the 14B model are substantial, particularly on the AIME25 benchmark.
> > 2. **Computational Overhead:** We clarified that the relevance calculation is pre-computed to avoid online overhead. We provided a detailed breakdown of the training step time and, most importantly, presented results showing that **LANPO outperforms a GRPO baseline that was trained for a comparable amount of wall-clock time**, confirming the gains come from improved sample efficiency, not just increased computation.
> > 3. **Relevance Filtering:** We clarified that the primary goal of the relevance filter is to enable the model to effectively leverage retrieved experiences at test time. Our ablation shows that without this filtering, test-time augmentation actually hurts performance, whereas our proposed method successfully improves it, demonstrating the filter's consistent utility for this core objective.
> >
> > We believe these additions and clarifications have significantly improved the paper. We have incorporated all new results and discussions into the revised manuscript.
> >
> > We would be very grateful to hear your thoughts on our response when you have a moment. We are eager to engage in further discussion and answer any additional questions you may have before the discussion period ends.
> >
> > Thank you once again for your valuable contribution to the review process.
> >
> > Best Regards,
> >
> > The Authors

---

### Author Response · Authors · 2025-11-30
**Summary of Review and Rebuttal**

Dear Program Chairs, Senior Area Chairs, Area Chairs, and Reviewers,

We sincerely appreciate the tremendous efforts of the Program Chairs, Senior Area Chairs, and especially Area Chairs in coordinating the review process. We also extend our gratitude to all Reviewers for their constructive reviews.

In light of the recent updates to OpenReview, we provide the following summary of the discussion phase to aid the Area Chair in navigating the rebuttal.

---

**Paper Summary:** Reinforcement learning for LLMs relies on scalar rewards, a practice that discards rich textual rationale buried in the rollouts and thus hampers sample efficiency. To address this, we propose Language-And-Numerical Policy Optimization (LANPO), a framework that introduces language feedback into the classic reward-driven reinforcement learning: It transforms on-policy rollouts into textual experiences to guide exploration, while using numerical rewards to drive optimization. We identify critical challenges in integrating these two feedback types and presents effective mechanisms to resolve them. When evaluated on 4B, 7B, and 14B models across math and coding benchmarks, LANPO achieves superior zero-shot accuracy over GRPO and incentivizes the development of novel reasoning skills like self-reflection and experience-based reasoning at test-time.

---

**Additional experiments.** The primary concerns about our initial submission centered on the experimental evaluation and computational overhead. In response, we have conducted additional experiments following the suggestions and updated the results into the manuscript:

1. Extended evaluation on coding benchmarks, now presented in Appendix A.5 (Reviewer yu24’s W2, Reviewer 7DfC’s W1/Q1).
2. Evaluation on a latest small LLM, Qwen3-4B-2507, now presented in Appendix A.4 (Reviewer yu24’s W1, Reviewer Wmpd’s W3).
3. Detailed computational overhead and comparison with GRPO under similar training time, now presented in Appendix A.6 (Reviewer yu24’s W3, Reviewer 7DfC’s W3/Q2, Reviewer Wmpd’s W2).

---

**Additional clarifications.** Besides experiments, we have provided additional discussions and clarifications to address the concerns of reviewers:

1. Necessity of our proposed Relevance Filtering, included in Line 455-462 (Reviewer yu24’s W4, Reviewer 7DfC’s W5)
2. Effectiveness of our proposed Reward-Agnostic Reflection (Reviewer 7DfC’s W4/Q4)
3. Training stability of LANPO and its on-policy characteristic (Reviewer 7DfC’s W3/Q3)
4. Forward/backward propagation when integrating language feedback, revised in Line 286-303  (Reviewer Wmpd’s Q1/Q2)

---

**Revised Writings.** We have also revised the writing and presentation of manuscript according to reviewers’ suggestions:

1. Caption of Figure 2, revised in Line 144-145 (Reviewer Wmpd’s W4)
2. Clearness of term choices, revised in Line 264-267 (Reviewer Wmpd’s Q3.1-Q3.3)
3. Formulation of LANPO, revised in Line 286-303  (Reviewer Wmpd’s Q3.4)

---

**Responses from Reviewers.** Two reviewers acknowledged the additional results and clarifications, and **one** **increased his ratings before Nov 27**:

1. Reviewer Wmpd notes our research as **“*a very interesting direction*”,** with potential to be **“*an important contribution to the research community*”.** He “*especially appreciate the empirical exploration*”, which “*provide valuable insights.*” He concludes that **our method “*effectively improves vanilla GRPO performance”.*** **His concerns “*are mostly addressed*”** by our response and comments our work as **“*a good contribution to the research community*”,** with a remaining problem on the new reasoning abilities. We have provided further explanations into the problem, but he did not reply before the update.
2. Reviewer 7DfC comments our research as **“*underexplored yet important*”**, believes our technical design as *“well-motivated components”,* and summarizes our experiments as ***“Consistent performance improvements”**.* According to himself, our initial response **address all his concerns except for Q3**, and **raised his rating on Nov 26**. We have provided further discussion about the only remaining concern, but he did not respond to us.
3. Reviewer yu24 acknowledges LANPO as a ***“novel and thoughtful way to incorporate language feedback into RL”.***  He highlights that *“LANPO achieves **better sample efficiency than GRPO**, while **improving both zero-shot and feedback augmented inference**”*, and our paper *“is well written with a number of insightful ablations”.* He did not respond to our rebuttal.

---

Overall, we are grateful for the insightful feedback, which has guided us in strengthening the manuscript. We have thoroughly addressed the points by incorporating new experiments and clarifications, and we are encouraged by the positive responses from reviewers who have re-engaged. We are confident these updates make our paper a stronger contribution to the community.

Thank you for your dedication.

Best regards,
Authors

---

### Meta-Review · Area_Chair_Uvce · 2026-01-08

**Summary:**

This paper proposes LANPO (Language-And-Numerical Policy Optimization), a framework that integrates language feedback into RL training for LLMs. The core contribution is identifying two failure modes when naively incorporating language feedback: information leakage and behavior collapse, and proposing Reward-Agnostic Reflection and Relevant Abstraction to address them.

The concerns were raised during review period including: (1) limited experimental scope (only Qwen models, primarily math tasks), (2) computational overhead without proper analysis, (3) unclear utility of the relevance filtering mechanism, (4) potential off-policy effects from prompt modifications, and (5) whether improvements stem from genuine reasoning capabilities or simply longer context.

Overall, reviewers recognized the contribution but were not strongly enthusiastic, resulting in borderline scores that lean toward rejection.

**Reviewer Concerns:**

During the rebuttal period, addressed concerns including: experiments on coding domain for the limited scope concern, detailed breakdown of the training step time for computational overhead, and clarification on Relevance Filtering. Outstanding concerns including the off-policy effects, rigorous evidence for new reasoning capabilities (from Reviewer Wmpd).

**Reviewer Scores:**

Reviewer yu24: 80% -> 6, 20% = 4. Reviewer 7DfC -> 4. Reviewer Wmpd = 6.

---

### Decision · Program_Chairs · 2026-01-26

Reject